

# Measurement report: Aircraft observations of aerosol and microphysical quantities of stratocumulus in autumn over Guangxi Province, China: Diurnal variation, vertical distribution and aerosol-cloud relationship

Sihan Liu[a], Honglei Wang[a*], Delong Zhao[b*], Wei Zhou[b], Yuanmou Du[b], Zhengguo Zhang[c], Peng Cheng[c], Tianliang Zhao[a], Yue Ke[a], Zihao Wu[a], Mengyu Huang[b]

[a] China Meteorological Administration Aerosol-Cloud and Precipitation Key Laboratory, Nanjing University of Information Science and Technology, Nanjing 210044, China

[b] Beijing Weather Modification Center, Beijing 100089, China

[c] Weather Modification Office of Guangxi Zhuang Autonomous Region, Nanning 530022, China

*Corresponding author.

*E-mail address:* hongleiwang@nuist.edu.cn (H. Wang) and zhaodelong@bj.cma.gov.cn(D. Zhao)

**Abstract:** Aerosols and clouds play important roles in the global climate system, and aerosol-cloud interactions have a significant impact on the radiation balance, water cycle, and energy cycle of the earth-atmosphere system. To understand the effect of aerosols on the vertical distribution of stratocumulus microphysical quantities in southwest China, the daily variation characteristics and formation mechanism of the vertical profiles of stratocumulus microphysical characteristics in this region were described by using the data of 9 cloud-crossing aircraft observations over Guangxi from October 10 to November 3, 2020. The influence of aerosol number concentration on cloud microphysical quantity was analyzed by combining the source of air mass and individual cases. Aerosol number concentration (Na) and cloud droplet concentration (Nc) both decreased gradually with the increase of altitude below 1500m, and did not change with the height between 1500 m and 3300 m. The inversion layer at the top of the boundary layer (PBL) hindered the increase in the cloud droplet particle size. The lower layer of the stratocumulus cloud in Guangxi was mainly small particle-size cloud droplet (effective diameter of cloud droplet, $E_d$<15 μm), and the middle and upper layer cloud droplet was large particle-size cloud droplet ($E_d$>20 μm). The vertical distribution of cloud microphysical quantity had apparent diurnal variation. When aerosols in the boundary layer





were transported to the upper air (14:00 to 20:00), the number of cloud droplets (Nc) in the lower
layer decreased, and the small particle-size cloud droplets ($E_d$ <20 μm) in the middle layer and upper
layer increased. When aerosols were transported to the boundary layer (10:00 to 13:00), the number
of small particle-size cloud droplets in the lower layer of the cloud increased. The characteristics of
cloud microphysical quantity were also affected by the source of air mass and the boundary layer.
Under the influence of land air mass or in the boundary layer, the aerosol number concentration (Na)
and Nc were high, and the cloud droplet number concentration spectrum was unimodal. Na and Nc
were low under the influence of marine air mass or above the boundary layer, and the cloud droplet
number concentration spectrum was bimodal. The relationship between stratocumulus and aerosol
in this region is consistent with the Twomey effect. $E_d$ and Na remain negatively correlated in
different liquid water content ranges, and FIE ranged from 0.07 to 0.58.
**Key words:** Aerosol; Aircraft observations; Cloud microphysical quantities; Vertical profile; the
boundary layer

## 1.Introduction

Clouds are an essential component of the Earth-atmosphere system, covering over 67% of the

Earth's surface (King et al., 2013), with stratocumulus clouds accounting for approximately 20% of
land and water surfaces (Wood, 2012). They can absorb atmospheric long-wave radiation and
reflecting solar short-wave radiation to influence the radiation budget of the Earth's atmospheric
system (Pyrina et al., 2015; Ramanathan et al., 1989; Zelinka et al., 2014). Additionally, they
participate in the global water cycle through precipitation processes (Betts, 2007; Rosenfeld et al.,
2014). Cloud microphysical characteristics are closely related to the climate effect and precipitation
formation of stratocumulus clouds. Differences in cloud water content, cloud droplet number
concentration and cloud droplet size in different regions will produce different radiative forcing and
precipitation (de Boer et al., 2008; Waliser et al., 2011; Yuan et al., 2008).

Aerosols are an important source of cloud condensation nuclei, and thus variations in aerosols

can lead to significant changes in the microscopic characteristics of clouds (Chen et al., 2021; Dusek
et al., 2006; Lance et al., 2004). Twomey(1977) suggested that, with the liquid water path of clouds
remaining constant, an increase of aerosol number concentration (Na) would increase in cloud
droplet number concentration (Nc) and a decrease in cloud droplet size, thereby enhancing cloud
albedo. Albrecht(1989) proposed that the decrease of cloud droplet particle size caused by the





increase of aerosols would further inhibit the precipitation process of clouds and thus extend the
lifetime of clouds.
Currently, aircraft observation, ground-based remote, and satellite remote sensing are the main
observation methods used to study the interaction between aerosol and cloud. Many scholars have
confirmed the Twomey effect through observational data (Ferek et al., 1998; Han et al., 1994;
Kleinman et al., 2012; Koren et al., 2005). Based on radar observation data, Kim et al.(2003) found
that AOD in Oklahoma presents a linear proportional relationship with liquid water path (LWP) on
a completely cloudy day with single-layer clouds, and the effective radius of cloud droplets is
negatively correlated with the surface aerosol light scattering coefficient. For a given LWP, Cloud
albedo and radiative forcing are very sensitive to the effective radius. Li et al.(2019) using aircraft
observation data over the Loess Plateau, found a negative correlation between Na and Nc in both
vertical and horizontal directions. Under high aerosol loading, smaller cloud droplets with higher
droplet number concentration were observed under high aerosol loading, while fewer larger cloud
droplets were formed under low aerosol loading. Cloud droplet number concentration was
negatively correlated with cloud droplet diameter within a certain range of liquid water content.
However, some scholars have also observed a positive correlation between aerosol and effective
diameter of cloud droplets ($E_d$) (Harikishan et al., 2016; Jose et al., 2020; Liu et al., 2020), referred
to as the anti-Twomey effect.
Aircraft observations with continuous vertical sampling are the most reliable source that can
accurately characterize the vertical relationship between aerosol and cloud (Nakajima et al., 2005;
Terai et al., 2014; Wehbe et al., 2021; Zaveri et al., 2022). McFarquhar et al.(2021) conducted
aircraft observations in the Southern Ocean region. They found aerosols above clouds may originate
from new particle formation and remote transport from continental air masses. This leads to cloud
condensation nuclei (CCN) variations and droplet concentration near cloud tops. During the ACE-
ENA campaign, the probability of aerosol transport interacting with marine boundary layer clouds
over the eastern North Atlantic (ENA) during summer was approximately 62.5% (Wang et al., 2020).
Zhao et al.(2019) observed a stratus cloud (water cloud) in the Huanghua region of China by
aircraft and found that in the planetary boundary layer (PBL) cloud, the effective radius of cloud
droplets and Na show a negative relationship, while they show a clear positive relationship in the
upper layer above PBL with much less Na. It also shows that the relationship between the effective



radius of cloud droplets and Na changes from negative to positive when LWC increases. Lu et
al.(2007) compared the microphysical quantities of stratocumulus clouds influenced by aircraft
flight tracks and those in undisturbed regions, and found that the effective radius of cloud droplets
in the flight path region was smaller, the number concentration of hair drops was lower, and the
cloud LWC was larger, providing observational evidence for the first indirect effect.
The mechanism of interaction between aerosols and clouds still involves significant uncertainty,
influenced by factors such as aerosol physicochemical properties, meteorological conditions, cloud
types, and the relative positioning of aerosols and cloud layers (Almeida et al., 2014; Dusek et al.,
2006; Wex et al., 2010; Zhang et al., 2011). Therefore, precise measurements of cloud microphysical
properties are crucial as the first step in studying aerosol-cloud interactions. Multi-aircraft
observations provide high-precision observational data, aiding in understanding the relationship
between aerosols and cloud microphysical characteristics.
Our study on the vertical distribution of aerosol in the Guangxi region found that the vertical
profile of Na in this region has prominent diurnal variation characteristics under the influence of the
boundary layer. In the morning, aerosols are mainly concentrated in the boundary layer. With the
development of the boundary layer and the enhancement of turbulent activity, the aerosols near the
ground are diluted in the afternoon, and some of the aerosols will be transmitted to more than 2 km
altitude. At night, the rapid decline of the boundary layer top will increase Na near the surface. At
the same time, some aerosols will stay above the boundary layer top, forming a high-concentration
aerosol layer (Liu et al., 2024). Previous studies have indicated that aerosols can have microphysical
properties of clouds and enhance indirect effects by entrainment into cloud tops when aerosol
particles settle or clouds deepen (Lu et al., 2018; Painemal et al., 2014). This study used data from
nine cloud-penetrating aircraft flights to investigate the vertical distribution and formation
mechanisms of cloud microphysical properties in stratocumulus clouds over the Guangxi region.
Additionally, we discussed the differences in the impact of aerosols from different sources on
cloud microphysical properties. We demonstrated that this region's correlation between aerosols and
clouds conforms to the Twomey effect. The ultimate goal is to provide observational constraints for
the simulation of aerosol radiative forcing in global climate models.
**2. Data and methodology**
**2.1. Aircraft data and reanalyze data**



The Beijing Weather Modification Office (BJWMO) provided the data for this study, and nine
flights of stratocumulus clouds and aerosols over Guangxi were conducted using the King Air 350
ER turbo aircraft. The aircraft is equipped with the Aircraft Integrated Meteorological Measurement
System (AIMS-20, Aven tech Inc., Canada), which provides meteorological elements such as
temperature (T) and relative humidity (RH) with a time resolution of 1 s. A passive cavity aerosol
spectrometer probe (PCASP-100X, DMT Inc, USA) was installed to provide aerosol number
concentrations in the particle size range of 0.11 to 3 μm, with a time resolution of 1s, particle size
accuracy of 20%, and concentration accuracy of 16%. The Fast Cloud Droplet Probe (FCDP, SPEC
Inc, USA) was used to observe the cloud droplet concentration, cloud particle concentration and
cloud particle size distribution. Its principle is to detect particles from 2 μm to 50 μm by using
forward scattering technology, with a time resolution of 1s. All instruments were calibrated before
observation. The detailed principles of the airborne instruments can be found in the following
studies (Collaud Coen et al., 2010; Strapp et al., 1992; Zhang et al., 2009).
**Table1** Flight information for the measurement campaign.

| Date | Take-off/landing time(Local time) | Cloud base/cloud top height (m) | Inside cloud $N_a$(cm⁻³) | $N_c$(cm⁻³) | LWC(g·m⁻³) | $E_d$(μm) |
|---|---|---|---|---|---|---|
| 20201010 | 11:53–15:50 | 1203-1652 | 355(157) | 586(328) | 0.45(0.30) | 12.25(1.92) |
| 20201011 | 14:26–17:53 | 1261-1542 | 636(290) | 529(350) | 0.19(0.14) | 9.45(1.30) |
| 20201025 | 09:34–12:58 | 1076-3298 | 9(31) | 38(35) | 0.18(0.15) | 26.96(9.80) |
| 20201026 | 09:53–13:29 | 1367-3146 | 5(19) | 35(27) | 0.10(0.09) | 21.86(8.77) |
| 20201028 | 14:05–17:27 | 1664-2729 | 239(229) | 354(502) | 0.45(0.43) | 16.90(9.54) |
| 20201029 | 10:05–13:33 | 516-3266 | 1402(569) | 396(289) | 0.17(0.16) | 9.86(2.54) |
| 20201101 | 18:17–22:06 | 1661-2715 | 333(170) | 199(80) | 0.35(0.17) | 17.93(4.71) |
| 20201102 | 14:04–17:41 | 696-3145 | 177(174) | 136(97) | 0.22(0.15) | 17.45(3.51) |
| 20201103 | 14:17–17:28 | 2021-2938 | 44(30) | 139(57) | 0.29(0.10) | 15.73(3.56) |

Detailed data of this aircraft observation activity, including observation date, time, cloud
thickness and microphysical quantities, are summarized in Table 1. Compared with aircraft
observation data in other regions, the average liquid water content (LWC) in Guangxi was higher,
5.33 times that in North China, and the average cloud droplet diameter was larger, 2.58 times that



in North China (Zhao et al., 2011). Compared with Marine Stratocumulus (Lu et al., 2011; Miles et
al., 2000), the stratospheric clouds in Guangxi had higher cloud base height and greater cloud
thickness. The cloud microphysical characteristics of Stratocumulus observed in this study are close
to those of previous observations. Compared with Stratocumulus (non-precipitation warm cloud)
over eastern China, the Nc, LWC and $E_d$ of Stratocumulus in Guangxi region were larger. According
to previous studies (Liu et al., 2024), there were no special weather processes in the upper air and
on the ground in Guangxi during the observation period, which ensured the quality of the data and
the universality of the conclusions.

The vertical pressure velocity (Pa·s-1) was obtained from MERRA2, with a spatial resolution

of 0.625°*0.5°and 42 layers, and a temporal resolution of 3 hours. The data from the first to the
twenty-third layers, corresponding to pressure altitudes from 1000 hPa to 200 hPa, were selected,
covering the maximum altitude of aircraft observations. An average calculation was performed to
obtain the vertical pressure velocity for the Guangxi region from 8:00 to 20:00 during the
observation period, reflecting the diurnal variation characteristics of vertical airflow above the
region. This dataset has been used in several studies (Ge et al., 2021; Kennedy et al., 2011; Painemal
et al., 2021).
**2.2. Data processing**

To ensure data quality, this study selected the data that met the following conditions and the

flight macro record as the in-cloud data: Nc$\geqslant$ 10 cm$^{-3}$, LWC$\geqslant$ 10$^{-3}$g m$^{-3}$ (Gunthe et al., 2009;
Zhang et al., 2011). The observation records show that the clouds during the observation period
were stratocumulus clouds (non-precipitation warm clouds). Therefore, this paper's aerosol and
cloud microphysical data met the following conditions: observation height $\leqslant$ 4000 m, T> 0 ℃.

The microphysical quantities such as Nc, LWC and $E_d$ are calculated from the cloud droplet

spectrum data detected by FCDP. The calculation formulas are as follows:

$$Nc = \sum n_i \qquad (1)$$

$$LWC = \sum \frac{4}{3}\pi r_i^3 \rho_w n_i \qquad (2)$$

$$E_d = 2\frac{\sum n_i r_i^3}{\sum n_i r_i^2} \qquad (3)$$

Define the relative height of the cloud as Zn



$$Zn = \frac{Z - Z_{base}}{Z_{top} - Z_{base}} \qquad (4)$$

$Z_{base}$ is the height of the cloud base, and $Z_{top}$ is the height of the cloud top

Similar to previous studies, the first indirect effect or Twomey effect of aerosols and clouds is

defined as
$$FIE = -\left( \frac{\Delta \ln E_d}{\Delta \ln \alpha} \right)_{LWC} \qquad (5)$$

Where is the amount of α aerosol, which can be used as aerosol optical depth (Feingold et al., 2001),
aerosol extinction coefficient (Feingold et al., 2003), cloud condensation nucleus concentration and
aerosol number concentration (Che et al., 2021; Zhao et al., 2012; Zhao et al., 2018). The value of
FIE may vary with variables representing the amount of aerosol.
**3. Results and discussion**
**3.1 Vertical distribution characteristics of cloud microphysical quantities**

The average value of the observation data of 9 sorties was calculated at the interval of 10 m

height, and the average vertical profiles of Na (interstitial aerosol), Na (out cloud), cloud
microphysical quantity and meteorological elements during the observation period was obtained
(Fig. 1). Na (interstitial aerosol) decreased gradually with height, and was affected by aerosols in
atmospheric environment. Below 1500 m, Na was high, with an average value of 749 cm$^{-3}$. From
1500 m to 3300 m, Na (interstitial aerosol) was low, with an average of 107 cm$^{-3}$. Nc decreased first
and then remained unchanged with the increase of height. Below 1500 m, there were more cloud
condensation nuclei in the atmosphere, Nc was high, with an average value of 407 cm$^{-3}$. Between
1500 and 3300 m, the Nc changed little with the increase of height, and the range of Nc is 10-200
cm$^{-3}$ (Fig. 1a).

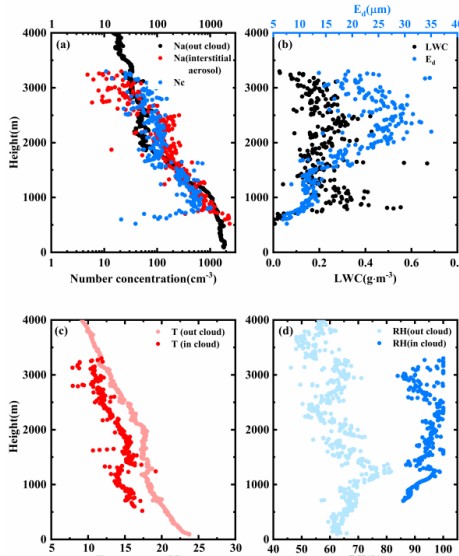

**Fig. 1** Average vertical profiles of cloud interstitial aerosol concentration, outside aerosol number concentration,
cloud droplet concentration (a), LWC, effective diameter of cloud droplet (b), temperature inside and outside cloud
(c), and relative humidity inside and outside cloud (d) during the observation period

With the increase of height, $E_d$ first increased, then remained unchanged and then increased

(Fig. 1b). A large number of cloud droplets competed for water vapor below 1500 m, which is not
conducive to the growth of cloud droplets, so the average Ed was only 11.21 μm. The temperature
inversion layer of 1000-1500 m (Fig. 1c) is an essential factor hindering the growth of cloud droplets.
Above 1500 m, Nc was lower than the near ground, and the lower atmospheric temperature was
conducive to increasing cloud droplet particle size. The average value of $E_d$ reached 22.78 μm. The
value of LWC was independent of height, with an average value of 0.22 g·m$^{-3}$ in Guangxi (Fig. 1b).
**3.2 Diurnal variation of the vertical distribution of cloud microphysical quantities**



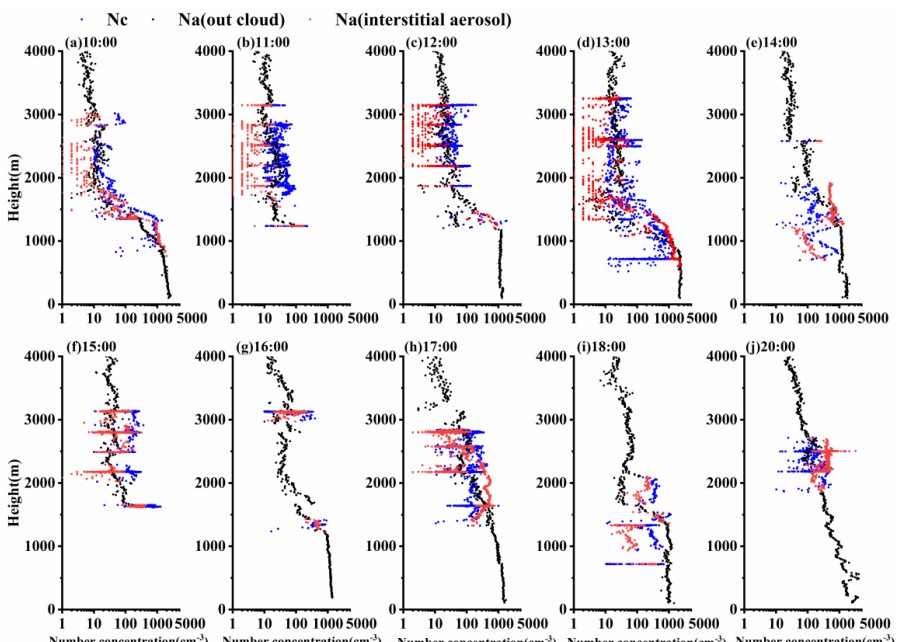

**Fig. 2** Vertical profiles of cloud interstitial aerosol concentration, outside aerosol number concentration, cloud droplet concentration at different times (a is 10:00, b is 11:00, c is 12:00, d is 13:00, e is 14:00, f is 15:00, g is 16:00, h is 17:00, i is 18:00, j is 20:00)

To further understand the diurnal variation of the vertical distribution of cloud microphysical quantities, the data were classified, and the vertical profiles of Na (interstitial aerosol), Na outside the cloud (Fig. 2), cloud microphysical quantities (Fig. 3), and meteorological elements inside and outside the cloud (Fig. 4) at 10 times from 10:00 to 18:00 and 20:00 were obtained. The data in the cloud was the original data, and the average value outside the cloud was calculated at the interval of 10 m.

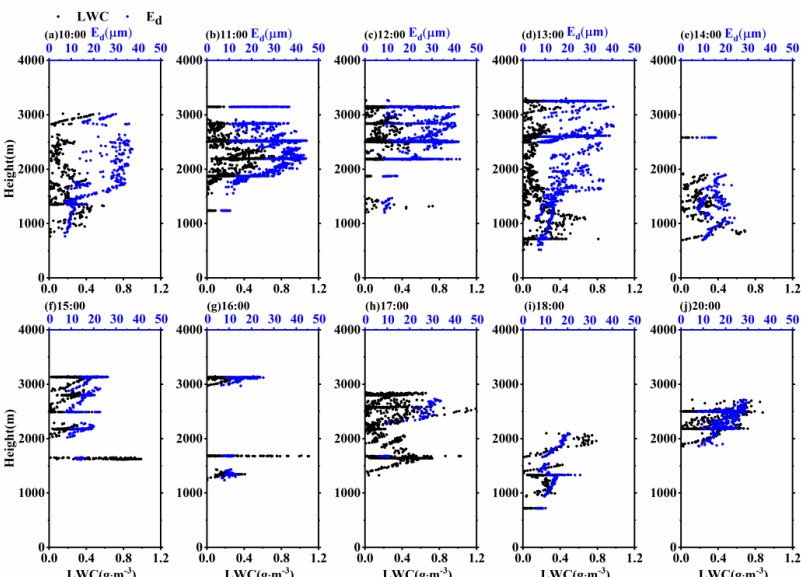

**Fig. 3** Vertical profiles of liquid water content and effective diameter of cloud droplets at different times (a 10:00,
b 11:00, c 12:00, d 13:00, e 14:00, f 15:00, g 16:00, h 17:00, i 18:00, j 20:00)

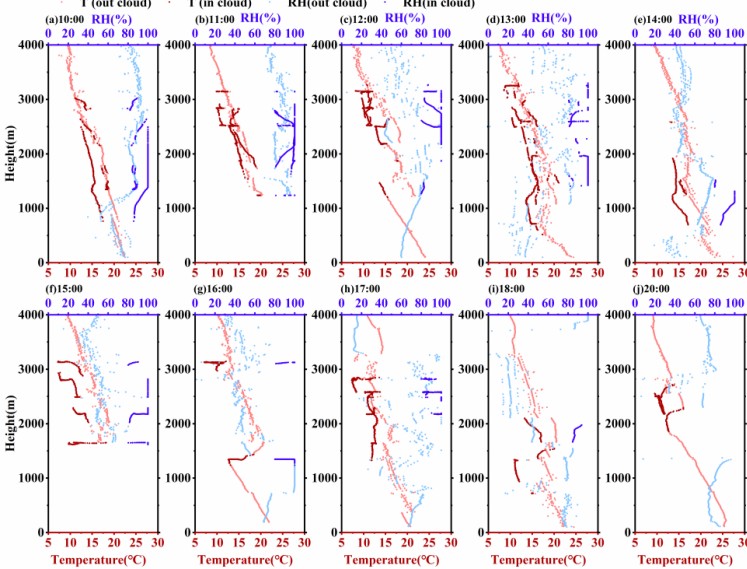

**Fig. 4** Vertical profiles of temperature inside and outside the cloud, relative humidity inside and outside the cloud
at different times (a is 10:00, b is 11:00, c is 12:00, d is 13:00, e is 14:00, f is 15:00, g is 16:00, h is 17:00, i is
18:00, j is 20:00)

At 10:00, Nc below 900 m was less than 100 cm$^{-3}$, and Na in PBL was high (Fig. 2a). Although

there were sufficient aerosols that can be activated into cloud condensation nuclei, the atmospheric





temperature was high, which was not conducive to the activation of aerosol particles (Fig. 4a). At
the same time, LWC was low, and the condensed cloud droplets are difficult to grow, and the average
$E_d$ is only 8.01 μm (Fig. 3a). Between 900m and 1500 m, there were not only sufficient cloud
condensation nuclei, but also sufficient water vapor and temperature conditions, which are
conducive to the formation of cloud droplets. The average Nc and $E_d$ increased to 430 cm$^{-3}$ and
11.15 μm. Above 1500 m, although the water vapor condition was sufficient, the cloud condensation
nucleus was few, resulting in an average Nc value of only 35 cm$^{-3}$. However, sufficient LWC was
conducive to the growth of cloud droplets, and $E_d$ was significantly higher than that of clouds below
1500 m, with Ed ranging from 13.82 to 37.26 μm. Na (interstitial aerosol) showed a slight increase
at 1500-1600 m, which may be due to the presence of a temperature inversion layer in the cloud
(Fig 4a), and the entanglement mixing at the cloud base mixed warm air outside the cloud into the
cloud (Lu et al., 2011).

At 11:00, more aerosols can activate into cloud condensation nuclei near the top of PBL. The

average Nc value was 102 cm$^{-3}$, while the average LWC value was only 0.03 g·m$^{-3}$, cloud droplets
were competing for water vapor, and $E_d$ was only 8.20 μm, which was similar to the cloud
microphysical characteristics near PBL at 10:00. From 1500 to 3150 m, Na was less than 10 cm$^{-3}$,
the cloud condensation nucleus was insufficient, and the average Nc was only 29 cm$^{-3}$. Compared
with 10:00, LWC was higher (mean 0.19 g·m$^{-3}$), so $E_d$ in the upper part of the cloud was more
significant, with an average of 28.95 μm

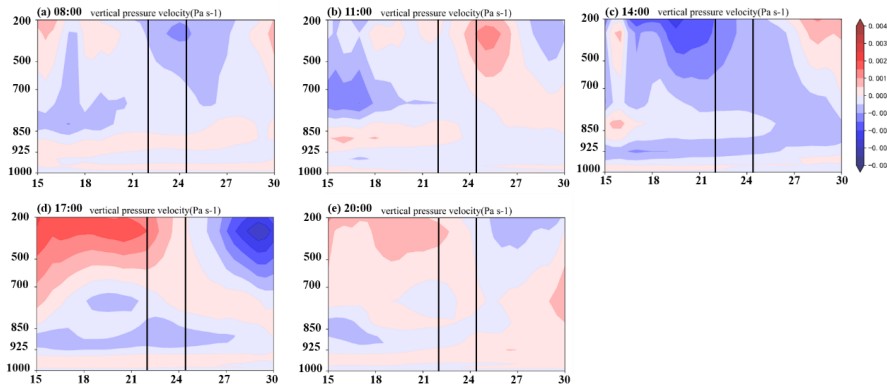


**Fig 5** Latitudinal profiles of vertical pressure velocity at different times in Guangxi (solid black line is the latitude
range observed by aircraft, positive value is downdraft, negative value is updraft, a is 8:00, b is 11:00, c is 14:00, d
is 17:00, e is 20:00)

At 12:00, under the combined action of the uplift of PBL and the updraft (Fig. 5b), the near-



surface aerosol was transported to 1200~1500 m (the mean value of Na outside the cloud was 578

cm$^{-3}$), the mean value of Nc reached 399 cm$^{-3}$, and the mean value of E$_d$ was only 9.41 μm, higher

than 11:00. Stratocumulus clouds above 1800 m have low Nc (mean 35 cm$^{-3}$) and large E$_d$ (mean

26.14 μm).

At 13:00, below 1200m, the range of Nc was 13~2052 cm$^{-3}$, possibly due to the different

degrees of cloud development at different locations. Compared with 10:00, the updraft speed in

Guangxi at 13:00 increased (Fig. 5b-c), which was conducive to the condensation of cloud droplets.

Therefore, Nc at 13:00 is more significant than that at 10:00, and many cloud droplets hinder their

particle size growth, with an average E$_d$ value of 9.23 μm. From 1200 m to 1500 m, the mean values

of Nc and E$_d$ were 155 cm$^{-3}$ and 12.29 μm. At this height, a strong inversion layer appeared and

cloud droplet evaporation activity was enhanced (Li et al., 2003), resulting in a higher Na (interstitial

aerosol) than Na (out cloud). For Stratocumulus clouds above 1500 m, the Nc varied little with

height, and the average E$_d$ was 21.45 μm.

At 14:00, the Nc range below 1500 m was 11-1109 cm$^{-3}$, and the top height of PBL was the

highest, which diluted aerosols in PBL. The average value of LWC was 0.29 g·m$^{-3}$ (Fig. 3e), which

was higher than 13:00, providing water vapor conditions for the growth of cloud droplets. Therefore,

the average value of E$_d$ reached 13.75 μm. An inversion layer appeared at 2500 m, hinding the

diffusion of aerosols and enhancing the evaporation of cloud droplets near the cloud top, resulting

in the peak of Na (interstitial aerosol) at this height.

At 15:00, there were more cloud droplets and cloud interstitial aerosol near the top of PBL,

and the average value of Nc and Na (interstitial aerosol) was 720 cm$^{-3}$ and 249 cm$^{-3}$ (Fig. 2f). The

average value of E$_d$ was only 13.72 μm due to more cloud droplets. The updraft transports low-level

aerosols upward, increasing the mean value of Nc above 2000 m to 146 cm$^{-3}$. The mean value of E$_d$

decreased to 16.73 μm due to the increase of small particle-size cloud droplets.

At 16:00, Nc and Na (interstitial aerosol) below 2000 m were relatively large, which were 458

and 468 cm$^{-3}$, respectively. The inversion layer at the top of PBL hinders the condensation growth

of cloud droplets, and the average E$_d$ was only 11.00 μm (Fig. 3g). Similar to 15:00, Na (out cloud)

and Na (interstitial aerosol) near 3000 m were higher. Low temperature and high humidity (Fig. 4g)

were conducive to the activation of aerosol, so the maximum value of Nc reaches 395 cm$^{-3}$, and the

average value of E$_d$ is only 17.13 μm due to water vapor contention between cloud droplets.



At 17:00, the height of the top of PBL decreased, and part of the aerosol remained above PBL,
providing enough cloud condensation nuclei. Nc did not change with the height (Fig 2h), with an
average value of 134 cm-3, and Ed concentrated in the 10-20 μm (Fig 3h). Under the cooling of the
atmosphere and the cooling of the cloud tops at sunset, the Ed near the cloud tops is greater than 30
μm. The temperature inversion layer of 1600~2000 m (Fig. 4h) enhanced the growth of cloud
droplets and impeded the diffusion of aerosols, resulting in Na (interstitial aerosol) being more
significant than Na (out cloud).

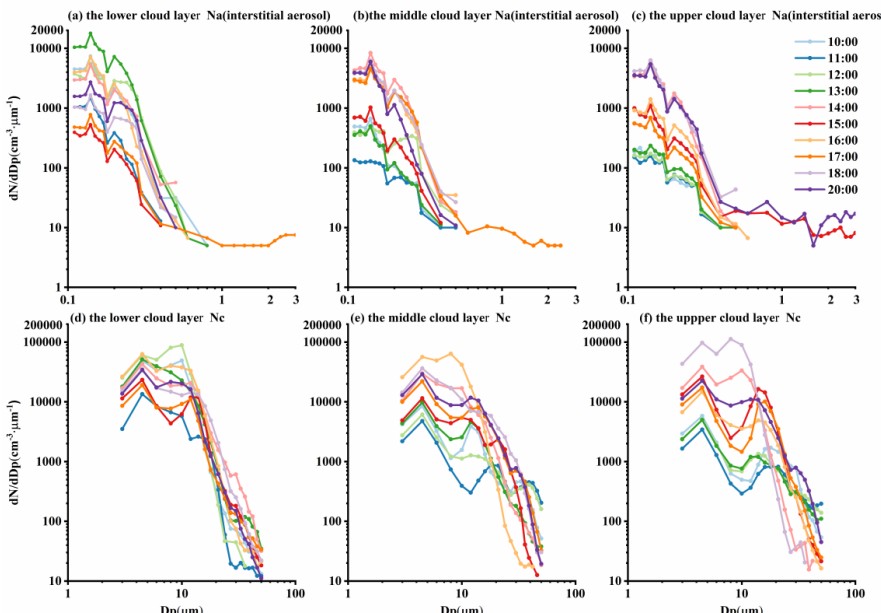


**Fig 6** Cloud interstitial aerosol number concentration spectrum and cloud droplet number concentration spectrum
(a-c is the aerosol spectrum of lower cloud, middle cloud and upper cloud, and d-f is the cloud droplet spectrum of
lower cloud, middle cloud and upper cloud, respectively)
At 18:00, the rapid decline in the top height of PBL resulted in the accumulation of aerosol in
the outer cloud between 900 m and 1400 m (Fig. 2i), which provided sufficient cloud condensation
nuclei and was conducive to the formation of more small particle-size cloud droplets (the mean Nc
value was 273 cm-3, and the mean $E_d$ value was 16.67 μm). Similar to 17:00, the atmospheric
ambient temperature above 1400 m was high, and cloud droplet evaporation caused Na (interstitial
aerosol) to be close to or greater than Na (out cloud).
At 20:00, the aerosols lingering above the top of PBL spread to the upper air with the updraft,
and sufficient cloud condensation nuclei and low temperature were conducive to the formation and





growth of cloud droplets. The mean value of Nc was 194 cm$^{-3}$ respectively (Fig. 2j), which was
much higher than Nc from 10:00 to 13:00. LWC and $E_d$ gradually increased with height (Fig. 3j).
LWC increased from 0.02 g·m$^{-3}$ to 0.64 g·m$^{-3}$, and $E_d$ increased from 7.52 μm to 29.59 μm.

The cloud height is normalized, and the relative height of the cloud is set as Zn (0≤ Zn≤ 1).

Zn< 0.33 is defined as the lower cloud layer, 0.33≤ Zn< 0.67 is the middle cloud layer, and Zn≥
0.67 is the upper cloud layer. The concentration spectra of cloud interstitial aerosol numbers (Fig.
6a-c) and cloud droplet numbers (Fig. 6d-f) at different locations at different times were obtained.
The vertical distribution of cloud microphysical quantities of stratocumulus in Guangxi showed
noticeable diurnal variation.

From 10:00 to 13:00, Na and Nc in the cloud lower layer were large, cloud droplet diameter

was concentrated below 20 μm, large cloud droplet was less, and $E_d$ was small. Na and Nc in the
middle cloud layer were less than those in the lower cloud layer. Na and Nc the upper cloud layer
are the smallest. Sufficient water vapor was conducive to the hygroscopic growth of cloud droplets,
so the number of small particle-size cloud droplets in the upper cloud layer was smaller than that in
the middle cloud layer.

From 14:00 to 16:00, the updraft diffused aerosols in the lower atmospheric layer upward, and

Na and Nc in the lower cloud layer decreased, which was conducive to the condensation growth of
cloud droplets, and the number of cloud droplets with a diameter greater than 20μm increased. The
upward transport of aerosol causes Na and Nc in the cloud and the upper layer to be larger than
those from 10:00 to 13:00, and newly generated cloud droplets compete for water vapor, resulting
in a decrease in the number of cloud droplets larger than 30 μm in diameter and an increase in small
particle-size cloud droplets.

From 17:00 to 20:00, the descending and downdraft of the boundary layer increase Na and Nc

in the lower cloud layer, and the number of large cloud droplets is less than 14:00 to 16:00. The
aerosol retained above the top of the boundary layer provides sufficient cloud condensation nuclei
for the middle layer and upper layer of the cloud. The Na and Nc in this period are higher than those
from 10:00 to 13:00, and the increase of cloud droplets with small particle-size is mainly due to the
increase of Na and Nc.
**3.3 Influence of aerosols on microphysical quantities of stratified clouds**

Previous studies have shown two sources of aerosols in Guangxi, namely the land and the



ocean, where air masses from land will bring higher aerosols. According to the classification of air
mass sources, the frequency distributions of Na (interstitial aerosol), Nc, LWC and $E_d$ under the
influence of land and ocean air masses were obtained (Fig. 7).

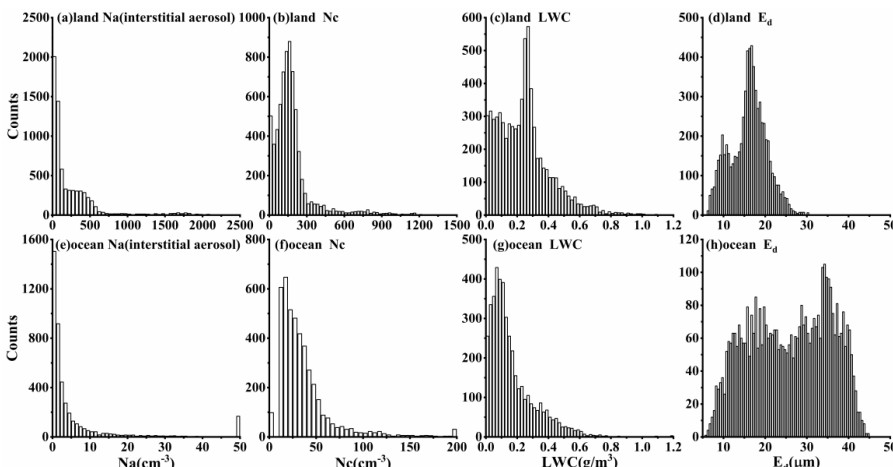

**Fig. 7** The distributions of cloud interstitial aerosol number concentration (a and e), cloud droplet number
concentration (b and f), LWC (c and g) and cloud droplet effective diameter (d and h) under different air mass
sources
Under the influence of land air mass, Na (interstitial aerosol) was less than 500 cm$^{-3}$, Nc was
high, and the frequency distribution of $E_d$ was unimodal, mainly concentrated in the range of 16-18
μm (Fig. 7a). Under the influence of ocean air mass, Na (interstitial aerosol) was mainly less than
20 cm$^{-3}$, and Nc was mostly distributed in the range of 10 to 50 cm$^{-3}$. $E_d$ was significantly higher
than that under the influence of land air mass. $E_d$ presented a bimodal distribution with peak values
of 17.75 and 34.25 μm (Fig. 7b).
In addition to the influence of the air mass source, the vertical distribution of Na is also affected
by PBL. We selected two aircraft observation data on October 29 and November 02 to analyze the
influence of PBL on the cloud microphysical quantities. The observed cloud base height was lower
than the PBL top's, and the cloud top height was> 1500 m. The clouds crossed the PBL top, and the
cloud thickness was similar (about 2500 m).
According to the vertical profiles of the aerosol number concentration spectrum (Fig. 8a-b),
there were significant differences between the two Na profiles. For the height affected by PBL
(below 1500 m), aerosol pollution occurred on October 29 (Na> 1000 cm$^{-3}$), and the atmosphere
was relatively clean on November 2 (Na< 600 cm$^{-3}$). For the upper atmosphere (above 1500 m),



aerosol pollution (Na< 200 cm$^{-3}$) occurred on November 2 compared to October 29 (Na< 100 cm$^{-3}$).

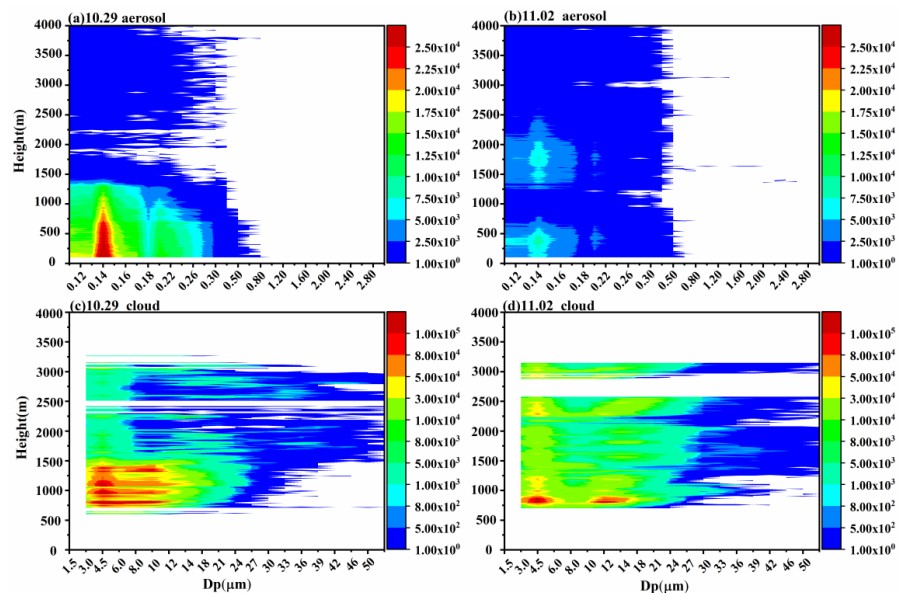

**Fig. 8** Vertical profiles of aerosol number concentration spectra (a and b) and cloud droplet number concentration

spectra (c and d) on 29 October and 2 November

On October 29, the aerosol pollution in PBL was severe (Na = 1331 cm$^{-3}$). The aerosol number
concentration spectrum was bimodal, with peak diameters of 0.14 and 0.22 μm. The atmosphere has
sufficient cloud condensation nuclei, so the Nc is large (Nc = 460 cm$^{-3}$). It can be seen from the
cloud droplet number concentration spectrum (Fig. 8c) that most of the cloud droplets were mainly
concentrated in the range of 3-24 μm. The average value of E$_d$ was only 9.69 μm, primarily because
many cloud droplets compete for water vapor and it was difficult for them to grow into large cloud
droplets. A strong inversion layer at 1500 m (Fig. 9c) made it difficult for aerosols to transmit
upward. Na above 1500 m was low, so Nc was also low, with an average value of only 35 μm. Fig.
8c showed that the concentration spectrum of cloud droplets presented a bimodal pattern, the
number of cloud droplets with particle size in the range of 8.0-21 μm decreased, and the particle
size of cloud droplets was mainly distributed below 8 μm and above 21 μm. The average E$_d$ reached
25.28 μm. These large particle-size cloud droplets may come from the collision and condensation
growth of cloud droplets in the range of 8.0 to 21 μm.

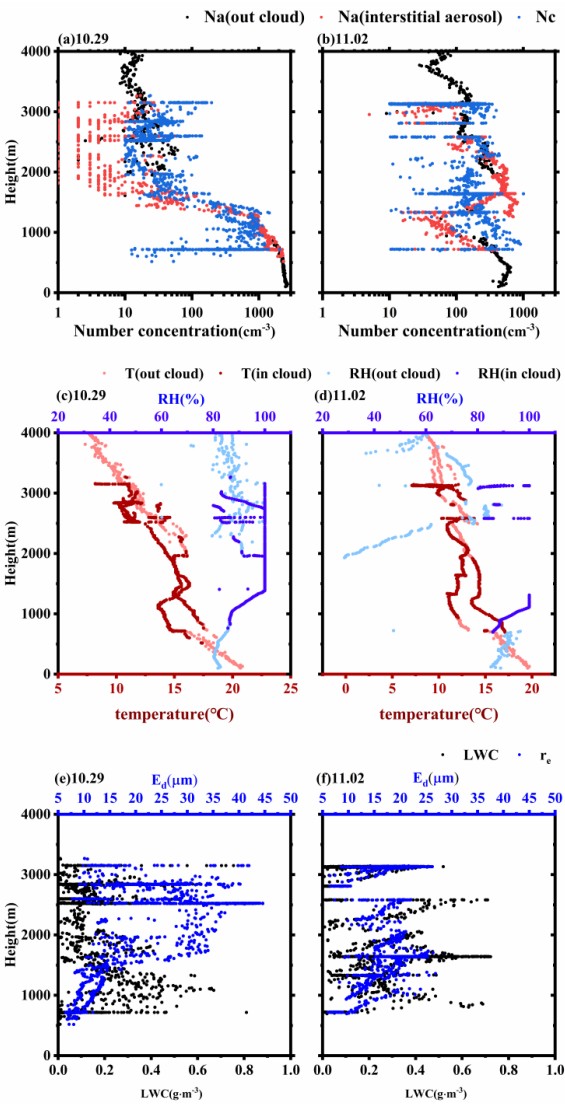

**Fig. 9** Vertical profiles of outside aerosol concentration, cloud intercloud aerosol concentration, cloud droplet concentration (a and b), temperature inside and outside the cloud, relative humidity inside and outside the cloud (c and d), LWC, and effective droplet diameters (e and f) on October 29 and November 2

On November 2, Na in PBL (mean 405 cm⁻³) was slightly higher than Na in the upper air (mean 220 cm⁻³). The aerosol spectral patterns were similar (Fig. 8b), both of which were bimodal and the peak particle sizes were 0.14 and 0.22 μm, indicating that the aerosol pollution over this area may come from the transportation of near-surface aerosols in the surrounding areas of Guangxi. Nc in PBL (mean 243 cm⁻³) was slightly higher than Nc above PBL (mean 124 cm⁻³). The concentration spectra of cloud droplet numbers were all bimodal (Fig. 8d). A large number of small particle-size




cloud droplets in PBL hinder the growth of large cloud droplets, so the number of large cloud
droplets (Dp> 18 μm) in PBL was lower than that in the upper air. And $E_d$ in PBL (mean 12.89 μm)
was lower than in the upper air (mean 17.94 μm). The inversion layer (about 750 m in thickness,
Fig. 9d) above the top of PBL enhanced the evaporation activity of cloud droplets, resulting in lower
Nc at this height than Nc at other heights, and higher Na (interstitial aerosol) than Na (interstitial
aerosol) at other heights.

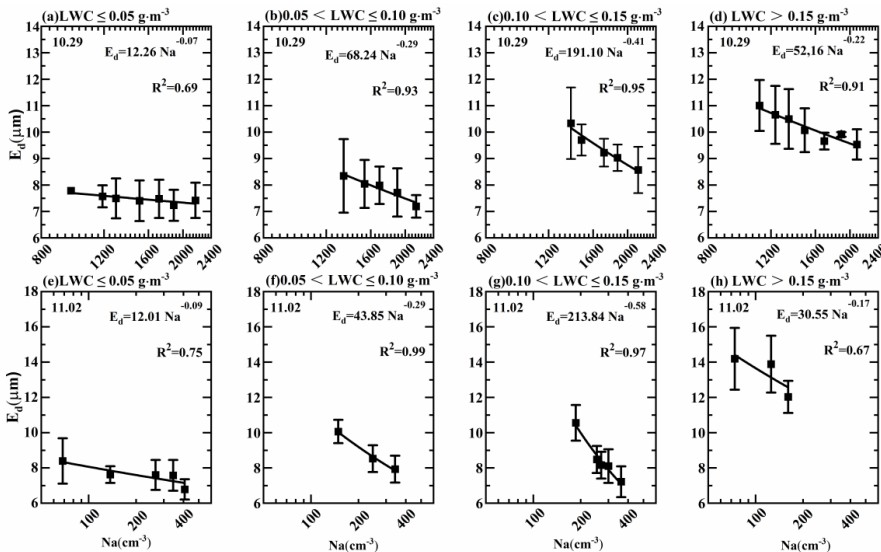


**Fig. 10** Correlation between aerosol number concentration and effective droplet diameter in the range of 0-0.05,
0.05-0.10, 0.10-0.15 and > 0.15 g·m⁻³ LWC (a-d are October 29, e-f are November 2, $R^2$ is the correlation
coefficient)
To understand whether the relationship between aerosol and cloud in Guangxi is consistent
with the Twomey effect, we classified the in-cloud data below 1000 m on October 29 and November
2. We calculated the FIE index of LWC in different ranges (Fig. 10). The results showed that Na
and $E_d$ were always negatively correlated regardless of low LWC condition or high LWC condition.
Therefore, the relationship between aerosol and stratocumulus in Guangxi is consistent with the
Twomey effect, and $E_d$ decreases with the increase of Na.
**4. Conclusion**
This study provides the vertical profiles of stratocumulus microphysical quantities, number
concentration spectrum and meteorological elements over Guangxi in autumn using the aircraft
observation data of 9 sorties. The diurnal variations of cloud microphysical characteristics at



different altitudes are described, and the effects of air mass source and the boundary layer on cloud
microphysical quantities are discussed. The results are as follows.
(1) Below 1500m in Guangxi, aerosol number concentration ($N_a$) and cloud droplet
concentration ($N_c$) both decreased gradually with the increase of altitude. Aerosol is mainly
concentrated near the ground, so $N_c$ was larger, with an average value of 407 cm$^{-3}$. Between 1500
m and 3300 m, $N_a$ was low, $N_c < 200$ cm$^{-3}$, and did not change with the height. With the increase in
height, the effective diameter of the cloud droplet ($E_d$) first increased, then remained unchanged,
and finally increased. $E_d$ of the cloud top was 2.75 times that of the cloud base. The inversion layer
at the top of the boundary layer (PBL) hindered the increase in the cloud droplet particle size.
Compared with other regions in China, LWC was high, with an average value of 0.22 g·m$^{-3}$, and
LWC variation was independent of height.
(2) The vertical distribution of microphysical quantities of stratospheric clouds in autumn in
this region had noticeable diurnal variation, mainly influenced by the diurnal variation of the vertical
distribution of aerosols. From 10:00 to 13:00, aerosols primarily concentrated in the low altitude,
so there were more small particle-size cloud droplets in the lower cloud layer ($N_c = 313$ cm$^{-3}$, $E_d =$
10.78 μm). From 14:00 to 16:00, under the combined action of lifting the top of PBL and the updraft,
the low-level aerosol diluted, decreasing the number of cloud droplets at the lower layer ($N_c = 184$
cm$^{-3}$). From 17:00 to 20:00, the descending and downdraft of PBL increased the number of small
cloud droplets at the lower layer ($E_d = 12.15$ μm). From 10:00 to 13:00, $N_c$ in the middle and upper
clouds was small, and the particle-size was large. From 14:00 to 20:00, the upward transport of
aerosols near the surface and the formation of a high concentration aerosol layer (600-1300 m)
increased the number of small particle-size cloud droplets in the middle and upper clouds.
(3) The source of air mass and PBL affected the distribution characteristics of cloud
microphysical quantities by influencing $N_a$. $N_c$ under the influence of the land air mass was 5.06
times that of the ocean air mass, but $E_d$ under the influence was 1.62 times that of the land air mass.
When there was a high number concentration of aerosol below PBL, the cloud droplet number
concentration spectrum was unimodal, and the cloud droplet size was concentrated below 24 μm.
Above PBL, the cloud droplet number concentration spectrum was bimodal, and the number of large
particle-size cloud droplets (cloud droplet diameter >27μm) was more than that in PBL. The
relationship between aerosol and cloud in the Guangxi region was consistent with the Twomey effect.



E$_d$ and Na were negatively correlated in different LWC ranges, and FIE ranged from 0.07 to 0.58.

**Competing interests.** The contact authors have declared that none of the authors has any competing
interests.

**Data availability.** All the aircraft data presented in this article can be accessed through
https://doi.org/10.5281/zenodo.13719678 (Wang, 2024). MERRA-2 data are available at
https://disc.gsfc.nasa.gov/daac-bin/FTPSubset2.pl (Bosilovich et al., 2015).

**Author contributions. SL**, **HW**, **DZ**, and **MH** designed this study. **WZ**, **YD**, **ZZ**, and **PC**
implemented the experiment and sample analysis. **SL** analysed the data and wrote the paper. **HW**,
**DZ**, and **TZ**: Funding acquisition, Writing - review & editing. **YK** and **ZW**: Data curation. All co-
authors proofread and commented on the paper.

**Acknowledgements.** The authors are grateful for the assistance of colleagues for sample
collection. We would like to thank the GMAO for MERRA-2 data
(https://disc.gsfc.nasa.gov/datasets?keywords=MERRA-2).

**Financial support.** This study was supported by the National Natural Science Foundation of
China (42075084), the National Key Research and Development Program of China (Grant No.,
2022YFC3701204) and the Natural Science Foundation of Jiangsu Province (BK20231300).

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
