# Peer review of "Measurement report: Aircraft observations of aerosol and"

_EGUsphere, 2024_

## Referee Comment (RC1)

Review of Manuscript EGUSphere-2024-2756:

**Measurement report: Aircraft observations of aerosol and microphysical quantities of stratocumulus in autumn over Guangxi Province, China: Diurnal variation, vertical distribution and aerosol-cloud relationship, by Liu et al.**

**General comments:**

Liu et al. presented aircraft observations on the physical properties of stratocumulus in autumn in 2020 over Guangxi Province in China, including the number concentrations of aerosol particles and cloud droplets, liquid water path, cloud droplet sizes, as well as cloud temperature and relative humidity. The authors first compared the above observations as a function of altitude to understand their vertical variabilities. In addition, the authors attempted to investigate the variabilities of the above observations at different times during the day from 10:00 to 20:00 hours (though the time zone was not provided). The authors further divided the stratocumulus, by using the so-called Zn values (defined by the equation 4), into three layers, including lower cloud layer ($0 \leq Zn < 0.33$), middle cloud layer ($0.33 \leq Zn < 0.67$), and upper cloud layer ($0.37 \leq Zn \leq 1.0$). Then, the authors compared the above cloud properties of each layer. Finally, the authors performed two case studies on Oct 29 and Nov 2, 2020. The authors also plotted the cloud droplet sizes as a function of the number concentration of aerosol particles for different cases with varying liquid water path and attempted to investigate the Twomey effect. Overall, I think I can follow the flow of the manuscript, though many sentences and statements are poorly and carelessly organized. Unfortunately, some figures (particularly Figures 5, 7 and 8) are poorly presented and some important results are lack of in-depth analysis. In general, significant amount of improvements need to be made carefully, regarding to results visualization, data interpretation, results discussion, and writing. Therefore, I contend that it should not be published in ACP as a measurement report.

I have seven major comments and many specific comments as below:

1. The writing of this manuscript has much space to be improved. It would benefit from a clearer thesis statement in many paragraphs and a more logical progression of concepts in many discussions. Please find my specific comments below and hope they could help. Before next submission to EGUsphere or elsewhere, a thorough proofreading or the assistance of a language editing service could help to get rid of many language issues.

2. A clear definition of stratocumulus in this study should be provided in the introduction. In addition, many important definitions are missing, including Zn, PBL (planetary boundary layer), PBL height, etc. Zn was introduced by equation 4. However, neither a relevant reference nor a further statement about more detailed physical explanation was provided. PBL hight, the concepts of inside and outside the PBL came up suddenly. How is PBL defined in this study? How did the authors determine the PBL height? All those were missing in the manuscript. Even relevant studies were hardly referenced appropriately.

3. This study only includes 9 aircraft observations during daytime between 10:00 and 20:00 hours. Nighttime observations are completely missing. Thus, the authors could not assert that the results from this study can present diurnal variabilities of stratocumulus clouds in the region.

4. In section 2.1, the number concentration of aerosol particles corresponds to particles between 0.11 and 3.0 μm, and the number concentration of cloud droplets have sizes between 2.0 and 50 μm. Does the size refer to optical size or aerodynamic size? If they are the same size with the same definition, there is an overlap between the two size ranges. Would the counted aerosol particles include droplets between 2.0 and 3.0 μm? or would the counted droplets include aerosol particles between 2.0 and 3.0 μm? How could this influence the results in this study? This was not evaluated/addressed in this manuscript.

5. Are the number concentrations of aerosol particles and cloud droplets in volumetric unit or are they already corrected to the standard atmospheric condition? The presented results in Figure 2 are from different days and different altitudes, they should be corrected to the standard atmospheric condition before being combined in one figure.

6. The authors stated the source apportionment of airmasses and tried to use the property difference between continental and marine aerosols to explain the differences in observed cloud properties. However, how was the source apportionment performed in this study? Or was it published in parallel studies? What are the physiochemical properties of these two kinds of airmasses? like size distribution and hygroscopicity which are important for acting as cloud condensation nuclei.

7. It is well known that aerosol particle number concentration and updraft velocity (i.e., water supersaturation conditions) are the two most important factors for the observed cloud droplet number concentrations. Although the authors presented the latitudinal profiles of vertical pressure velocity at different times (Figure 5), the discussions on the single effect of updraft velocity and the combination effects of these two factors are still lacking. The results in Figure 5a at 08:00 are irrelevant to results in Figures 2, 3 and 4, whereas the required results at 10:00, 12:00, 13:00, 15:00, 16:00 and 18:00 are missing. Also, the x-axis in Figure 5 is not labelled, it is neither indicated in the caption nor clarified in the text. The y-axis might have an unit in Pascal.

**Specific comments:**

Title: better to have 'aerosol-cloud interactions' rather than 'aerosol-cloud relationship'

Line 24: 'boundary layer (PBL)' or 'planetary boundary layer (PBL)'?

Line 25-26: I am confused about the sentence 'The lower layer of the stratocumulus cloud in Guangxi was mainly small particle-size cloud droplet' and also the phrase 'small particle-size cloud droplet'. How could the layer be cloud droplets?

Did you intend to state that the layer mainly contain small-sized cloud droplets? Or did you intend to state that the layer mainly contain cloud droplets which are formed by small-sized aerosol particles (i.e. small-sized cloud condensation nuclei)?

Line 28: Does "cloud microphysical quantity" refer to $N_c$ and $E_d$?

Line 29: Didn't you define $N_c$ already in Line 22?

Line 32: Did you mean the lower layer of the PBL will show increases in small-sized cloud droplets? It would be good to increase the clarity of the statement.

Line 34: $N_a$ was defined in Line 22 already.

Line 39: What does 'FIE' mean?

Line 40: It should be 'The boundary layer' or 'The planetary boundary layer'

Line 44-45: Stratocumulus clouds cannot account for land or water surfaces but can account for clouds that are over land or water surfaces.

Line 56-57: Please correct the grammar in the statement "… an increase of aerosol number concentration (Na) would increase in cloud droplet number concentration (Nc) and a decrease in cloud droplet size, …"

Line 61: Change "ground-based remote" to "ground-based remote sensing"

Line 62: Change "between aerosol and cloud" to "between aerosols and clouds"

Line 65: What is 'AOD'?

Line 67: should be 'cloud'

Line 70-72: In the sentence "Under high aerosol loading, smaller cloud droplets with higher droplet number concentration were observed under high aerosol loading, while fewer larger cloud droplets were formed under low aerosol loading.", how did you define low and high aerosol loadings? How do you compare higher and fewer cloud droplets? Numbers should be provided from the literature.

Line 72: Didn't you already define 'Cloud droplet number concentration' as '$N_c$' in Line 57? And in Line 73, did you mean liquid water path (LWP defined in Line 65) by saying 'liquid water content'. It is not far away. The same problem was shown in the abstract. More carefulness in writing will be appreciated. No further comments will be provided on such

abbreviations, including but not limited to $N_c$, $N_d$, $E_d$, LWP, and PBL. Please check through the whole manuscript. Detailed writing tips/rules can be found in ACP guidelines (via https://www.atmospheric-chemistry-and-physics.net/submission.html).

Line 74: Is it between aerosol number concentrations and the effective diameter of cloud droplets ($E_d$)? Or is it between one of any other aerosol properties and the $E_d$?

Line 77: Please refer to an exact study that directly compared aircraft observations with other methodologies and successfully demonstrated that aircraft observations are the most reliable method to investigate the vertical relationships between aerosols and clouds.

Line 81-82: Please correct the grammar in 'This leads to cloud condensation nuclei (CCN) variations and droplet concentration near cloud tops.'

Line 87: It should be 'they showed'.

Line 89: What is 'LWC'?

Line 92: What are 'hair drops'?

Line 93: The first indirect effect of what? Or it refers to Twomey effect? It is not clear and the readability should be improved.

Line 106-108: Is the statement relevant for your findings or are those findings are from Liu et al. 2024? "

Line 108-109: The statement of 'aerosols can have microphysical properties of clouds' sounds very wired.

Line 114-115: 'We demonstrated that this region's correlation between aerosols and clouds conforms to the Twomey effect.' The structure and grammar of this sentence can be much better developed.

Line 126: Is it 'accuracy' or 'uncertainty'?

Line 134-137: Average values should be provided.

Line 139-140: This statement contradicts the other statements. How do you mean by saying 'close to'? Does it mean 'similar to'? More rigorous and clearer logic to develop statements would be appreciated.

Line 140-141: Why to have capitalized 'Stratocumulus'?

Line 150: The observations from 08:00 to 20:00 hours do not represent diurnal observations. Also, relevant statements/conclusions on diurnal variations do not make sense based on daytime observations only.

Line 155: The LWC$\geq 10^{-3}$ g m$^{-3}$ is ambiguous.

Line 158: Is the observation height above seal level or ground level?

Line 161-163: What are $n_i$, $r_i$, and $\rho_w$?

Line 166: A full stop is missing.

Line 170: 'Where is the amount of α aerosol', this statement does not make sense.

Line 176-179: Sentence too long and wrong grammar. Also, how cloud the same $N_a$ represent two parameters of two cases (interstitial aerosols and out-cloud aerosols)? How did you calculate the average values? Is it calculated at the same height of nine flights? Or is it calculated as the average of observations in the height interval (10 m) for each flight and then you present the results if nine flights together? This should be clearly introduced in the manuscript.

Line 180-181: It is very hard to read these two average values in Figure 1a.

Line 181-185: The authors contradict themselves. They first stated that 'Nc decreased first and then remained unchanged with the increase of height.' On the contrary, they stated that 'Between 1500 and 3300 m, the Nc changed little with the increase of height, and the range of Nc is 10-200 cm-3 (Fig. 1a).'

Line 190: Altitude values for each stage should be specified.

Line 192-193: The reason why there exists an inversion layer for temperatures should be shortly but clearly explained. Why such a temperature inversion layer hamper cloud droplet growth should be elaborated.

Line 197: The results presented in this study cannot represent diurnal variations since the night results are missing.

Line 202-205: Sentence too long and with a poor structure.

Line 215: Is Nc less than 100 cm$^{-3}$ or is it lower than the detection limit of the instrument? What is the detection limit of the instrument? There are only a few data point for altitudes below 900 m.

Line 217: Figure 4a was mentioned earlier than Figure 3. So, why not switch these two figures.

Line 217: Why only temperature only is the reason? Is RH irrelevant? are the aerosol size and other physiochemical properties irrelevant?

Line 222: Which part of the results shows that water vapor is sufficient?

Line 225-228: Poor sentence structure. Is there any statistical results or results from Nc that could support this statement

stating that the temperature inversion layer and the increase in Na are correlated? Between 1500-1600 m, there are approximately 10 data points only.

Line 229: Where is the top of PBL? How was the PBL height determined/calculated?

Line 229-232: Very poor sentence structure.

Line 240-242: Can the pressure be calculated into altitude values? This way, it will be easier to compare to the results in Figures 2, 3 and 4.

Line 246: What do 'different locations' mean? Air mass regions? Or vertical locations?

Line 247: Figure 5b is for 11:00 and Figure 5c is for 14:00. How could these two panels be used for discussions on 13:00?

Line 248: what does 'more significant' mean? Is there a difference between those two beyond the uncertainty? It is also very ambiguous too say 'many cloud droplets', how many? From where one can read the quantity of the cloud droplets mentioned by the authors?

Line 250: A strong inversion layer of what?

Line 254: What does 'the top height of PBL was the highest' mean? Can the height of the PBL be compared to Nc number concentrations? From where, it presents the results of PBL height?

Line 257: An inversion layer of what?

Line 260: more cloud droplets than what?

Line 262-263: There are no results in Figure 5 or other figures to show the updraft. This interpretation is not groundless.

Line 263-264: Based on results from which Figure could the authors draw such a statement? Please specify.

Line 268: How do authors define low temperatures and high humidities? The development of the statement should be clarified.

Line 271: How could it be known that 'At 17:00, the height of the top of PBL decreased, and part of the aerosol remained above PBL,….'?

Line 273: What does 'Ed concentrated in the 10-20 μm' mean? given that $E_d$ is defined as a size.

Line 277: The same comment on the statement '… resulting in Na (interstitial aerosol) being more significant than Na (out cloud)' as the comment on Line 248.

Line 299: 'Na and Nc in the cloud lower layer were large', how large they are? And the results refer to which figure and which panel?

Line 299-300: It is very awkward to have 'Na and Nc in the cloud lower layer were large' and 'large cloud droplet was less'.

Line 301-302: 'Na and Nc the upper cloud layer are the smallest', should be in the upper cloud layer.

Line 302: How do the authors know the water vapor is sufficient?

Line 305: 'the updraft diffused aerosols in the lower atmospheric layer upward' read awkward. Better wording is needed.

Line 305-307: 'From 14:00 to 16:00, the updraft diffused aerosols in the lower atmospheric layer upward, and Na and Nc in the lower cloud layer decreased, which was conducive to the condensation growth of cloud droplets, and the number of cloud droplets with a diameter greater than 20μm increased.' One cannot understand what is the cause and what is the effect in the statement. This sentence is also very poorly organized.

Line 312: Which part of the results in this manuscript shows the downdraft of the PBL between 17:00 and 20:00?

Line 316-317: What is the minor reasons? Based on which kind of comparisons, can the authors be sure the major reason?

Line 318: What are stratified clouds? Are they the same as or different from the aforementioned stratocumulus? No further information can be found in the following text.

Line 319: Which previous studies? Any references? Are the airmass source apportionment in those studies relevant for the same period of observations in this study?

Line 320: What does it mean by stating that 'air masses from land will bring higher aerosols'? Higher aerosol particle number concentrations? Or aerosols in higher altitudes?

Line 319-337: Without a clear definition for y-axis in Figure 7, the corresponding discussions do not make sense.

Line 338-369: It is unbelievable to discuss the results in Figure 8 without a unit for the colormap.

Line 365-367: 'The aerosol spectral patterns were similar', similar to what? It is nonsense to do aerosol source apportionment in such a baseless way. Did the authors analyse the physiochemical properties of those aerosols samples and compare the results to those of aerosol pollution sampled in surrounding areas of Guangxi? Or did the authors conduct simulation experiments (e.g. HYSPLIT or FLEXPART) to calculate the trajectory of these airmasses? Even if the source apportionment is based on previously relevant studies, at least, the relevant publications should be cited.

Line 367-368: Again how can one read the PBL height in Figure 8?

Line 383: What is the significance level of the calculated correlation coefficient?

Line 388: meteorological parameters instead of 'meteorological elements'

Line 389: The results in this study are not diurnal since night time observations are missing.

Line 394-395: 'Between 1500 m and 3300 m, Na was low, Nc< 200 cm-3, and did not change with the height.' This is not academic writing.

Line 401-403: Again, the results are not diurnal.

**Figures and Tables:**

Table 1:  What do the values in brackets mean? Are these concentration values corrected to standard conditions?

Figure 1: The results in Figure 1d were not discussed or referred in the text. I am wondering the point of discussing the average values since more specific results as a function of altitude were provided in Figure 1. Given the large spread of the results in Figure 1, an average value of a selected range does not really make sense.

Figure 2: What is the time zone of the hours? Why the results at 19:00 is missing?

Figure 3: The same comments as those on Figure 2. The legend and axis labels can be better organized to improve its readability.

Figure 4: The same comments as those on Figure 3.

Figure 5: What is the time zone of the hours? What does the x-axis mean?

Figure 7: What does the y-axis (Counts) mean? There is no definition in the manuscript.

Figure 8: The unit for the colour map scale is missing.

Figure 10: Which kind of correlation coefficient is presented in this Figure? How was it calculated? What is the significance level ($p$ value) of the correlation coefficient? The maximum n value is only 7. Is the equation in each panel representing the function to calculate the correlation coefficient?

---

## Author Comment (AC1)

The manuscript presents a comprehensive investigation into the diurnal variation, vertical distribution, and aerosol-cloud relationships of stratocumulus clouds over Guangxi Province, China, utilizing aircraft observations. It offers insights into the impact of aerosol number concentration on cloud microphysical quantities, including cloud droplet concentration and effective diameter. The study reveals the interaction between the vertical structure of aerosols and the planetary boundary layer, highlighting the critical role of aerosols in influencing cloud microphysical characteristics. Given that China, particularly the region of Guangxi, is known for its high aerosol concentrations, this research contributes to the regional understanding of aerosol-cloud interactions, a topic that has been understudied in southern China.

The manuscript offers a detailed exploration of the diurnal variation, vertical distribution, and aerosol-cloud relationships of stratocumulus clouds over Guangxi Province, China, with a clear articulation of results and conclusions. To strengthen the manuscript, the authors should discuss the comparability of their findings with other regions globally, especially considering the unique meteorological conditions of Guangxi. At least add some discussion on this point. Enhancing the description of aircraft observation methods, including instrument calibration and data validation, is essential for credibility. Additionally, a thorough explanation of the statistical methods, including the choice of tests, significance levels, and assumptions, is necessary for the reader to replicate your results. Deepening the comparison with the existing literature\data report would provide better contextualization of the study's contributions. Further exploration of the physical mechanisms behind aerosol-cloud interactions would enrich the paper. Finally, outlining the possible implications for climate modeling, weather forecasting, and air quality management, along with suggestions for future research, would make the manuscript more impactful. Addressing these aspects will not only bolster the study's rigor but also its relevance to the broader scientific community.

Considering the above points, I recommend that the manuscript be accepted for publication in Atmospheric Chemistry and Physics (ACP) after the authors have addressed these revisions. The study's findings have the potential to significantly

contribute to the field of atmospheric science, particularly in understanding the complex dynamics of aerosol-cloud interactions in regions with high aerosol loading. But overall, I cannot require a data measurement report with the same requirements as an article, therefore I recommend a minor revision.

Dear reviewer:

Thank you for your decision and constructive comments on our manuscript. We have carefully considered the suggestions, tried our best to improve and made some changes in the manuscript. We have enhanced the discussion by comparing our results with studies from other regions to highlight the unique meteorological conditions of Guangxi. We have provided a more comprehensive description of our aircraft observation methods, including instrument calibration and data validation, to bolster the credibility of our findings. We have also clarified the statistical methods used in our analysis, detailing the tests' significance levels. We have deepened the comparison with existing literature and explored the physical mechanisms underlying aerosol-cloud interactions in more detail. We believe addressing these points will significantly enhance the manuscript's rigor and relevance. We appreciate your recommendation and are committed to implementing the necessary changes. Thank you once again for your valuable insights.

The blue part has been revised according to your comments. The line numbers are in the revised manuscript; the changes are identified in red. Revision notes, point-to-point, are given as follows:

Specific comments:

1. interstitial aerosol or interstitial aerosol particles, you didn't provide any information on it, which makes it hard to understand for the reader who is not familiar with this concept, (interstitial aerosol particles: particles too small to activate to cloud droplets)

Response: Thank you for your comment. We have added an explanation of gap aerosol particles in lines 193-196 to improve the manuscript's readability.

The average vertical profiles of Na (interstitial aerosol, aerosol particles too small to activate to cloud droplets), Na (out cloud) (Fig. 1a), cloud microphysical quantities

(Fig. 1b) and meteorological elements (Fig. 1c-d) during the observation period was obtained.

2. line 245, Nc was 13~2052, should be 13-2052 cm$^{-3}$, please don't use '~'.

Response: Thank you for your comment. We have carefully reviewed and corrected the manuscript's formatting errors.

3. The manuscript comprehensively analyzes the vertical distribution of aerosols and cloud microphysical quantities in stratocumulus clouds over Guangxi Province, China. The findings contribute to the understanding of interactions between aerosols and clouds. However, the authors should consider providing a more detailed comparison of their results with previous studies, especially those conducted in other regions with different meteological conditions.

Response: Thank you for your comment. We have added a comparative analysis of the vertical distribution of cloud microphysical properties with other regions to highlight the unique cloud microphysical characteristics of Guangxi Province.

Compared with aircraft observation data in other regions, the average liquid water content (LWC) in Guangxi was higher, 5.33 times that in North China, and the average cloud droplet diameter was larger, 2.58 times that in North China (Zhao et al., 2011). Compared with the Marine Stratocumulus (Lu et al., 2011; Miles et al., 2000), the Stratocumulus in Guangxi had higher cloud base height and greater cloud thickness. The cloud microphysical characteristics of the stratocumulus observed in this study are similar to those of previous observations. Compared with stratocumulus (non-precipitation warm cloud) over eastern China, the Nc, LWC and $E_d$ of stratocumulus in Guangxi region were larger.


Response:   Thank you for your comment. We have added the characteristics of

spectral distribution changes over time and analyzed the temporal variations in the boundary layer and the effects of meteorological factors on cloud droplet and aerosol spectra.

From 10:00 to 13:00, the interstitial aerosol particle size in the cloud's lower layer was concentrated below 0.4 μm. In comparison, the cloud droplet diameter was primarily concentrated below 20 μm, with few large particle-size cloud droplets (Fig. 6a, 6d). In the middle cloud layer, Na across all particle size ranges had decreased to below 1000 $cm^{-3} \cdot \mu m^{-1}$. Nc for particles smaller than 20 μm has decreased, while Nc for particles larger than 20 μm exceeded 0.1 $cm^{-3} \cdot \mu m^{-1}$ (Fig. 6b, 6e). Na in the upper cloud layer was minimal compared to the middle and lower layers. Sufficient water vapor (LWC = 0.14 $g \cdot m^{-3}$, Fig.3a-c) and low temperature (T = 11.72 °C, Fig.4a-c) promote the growth of cloud droplets, resulting in fewer Nc for particles larger than 20 μm in the upper layer (Fig. 6c, 6f) compared to the middle layer.

From 14:00 to 16:00, aerosols diffused upward with the increase in PBL, leading to a decrease in Na in the cloud's lower layer (Fig. 6a, 6d). The upward transport of aerosols caused the upper-level Na of the cloud to be higher than that observed from 10:00 to 13:00. This change increased the Nc of droplets with diameters greater than 20 μm (Fig. 6b-c, 6e-f). Newly formed cloud droplets competed for water vapor. Nc of droplets larger than 30 μm decreased, while Nc of smaller droplets increased.

From 17:00 to 20:00, the height of PBL decreased. Na increased, and Nc of large droplets decreased. Aerosols retained at the top of PBL provided CCN for the cloud's middle and upper layers (Fig. 6b-c, 6e-f). During this period, Nc was higher than observed from 10:00 to 13:00. The increase in Nc may be attributed to the rise in Nc of droplets smaller than 20 μm.

In conclusion, our findings highlight the significant influence of aerosol concentrations and air mass origins on the microphysical properties of stratocumulus clouds over Guangxi. The observed temporal and vertical variations in cloud microphysics underscore the complexity of aerosol-cloud interactions in this region. Future research should cover a comprehensive time frame, including nighttime

observations, to provide a complete vertical structure of these clouds, the effects of different aerosol types, and their impact on regional climate patterns.

---

## Author Comment (AC2)

General comments:

Liu et al. presented aircraft observations on the physical properties of stratocumulus in autumn in 2020 over Guangxi Province in China, including the number concentrations of aerosol particles and cloud droplets, liquid water path, cloud droplet sizes, as well as cloud temperature and relative humidity. The authors first compared the above observations as a function of altitude to understand their vertical variabilities. In addition, the authors attempted to investigate the variabilities of the above observations at different times during the day from 10:00 to 20:00 hours (though the time zone was not provided). The authors further divided the stratocumulus, by using the so-called $Z_n$ values (defined by the equation 4), into three layers, including lower cloud layer ($0 \leq Z_n < 0.33$), middle cloud layer ($0.33 \leq Z_n < 0.67$), and upper cloud layer ($0.37 \leq Z_n \leq 1.0$). Then, the authors compared the above cloud properties of each layer. Finally, the authors performed two case studies on Oct 29 and Nov 2, 2020. The authors also plotted the cloud droplet sizes as a function of the number concentration of aerosol particles for different cases with varying liquid water path and attempted to investigate the Twomey effect. Overall, I think I can follow the flow of the manuscript, though many sentences and statements are poorly and carelessly organized. Unfortunately, some figures (particularly Figures 5, 7 and 8) are poorly presented and some important results are lack of in-depth analysis. In general, significant amount of improvements need to be made carefully, regarding to results visualization, data interpretation, results discussion, and writing. Therefore, I contend that it should not be published in ACP as a measurement report.

I have seven major comments and many specific comments as below:

1. The writing of this manuscript has much space to be improved. It would benefit from a clearer thesis statement in many paragraphs and a more logical progression of concepts in many discussions. Please find my specific comments below and hope they could help. Before next submission to EGUsphere or elsewhere, a thorough proofreading or the assistance of a language editing service could help to get rid of many language issues.

Response: Thank you for your decision and constructive comments on our manuscript. We have carefully considered the suggestion and tried our best to improve and make changes to the manuscript. Meanwhile, we found a professional company to polish the language of this article again.

The blue part has been revised according to your comments. The line numbers are in the revised manuscript; the changes are identified in red. Revision notes, point-to-point, are given as follows:

2. A clear definition of stratocumulus in this study should be provided in the introduction. In addition, many important definitions are missing, including Zn, PBL (planetary boundary layer), PBL height, etc. Zn was introduced by equation 4. However, neither a relevant reference nor a further statement about more detailed physical explanation was provided. PBL height, the concepts of inside and outside the PBL came up suddenly. How is PBL defined in this study? How did the authors determine the PBL height? All those were missing in the manuscript. Even relevant studies were hardly referenced appropriately.

Response: Thank you for your comment. We have ensured that a clear definition of stratocumulus, as well as comprehensive explanations of terms such as Zn, PBL, and PBL height, are included in the introduction. We have incorporated the necessary information and references in the revised manuscript to address these omissions.

We supplement the introduction of stratocumulus and planetary boundary layer in lines 44-47.

Clouds are an essential component of the Earth-atmosphere system, covering over 67 % of the Earth's surface (King et al., 2013), with stratocumulus clouds covering approximately 20 % of the Earth's surface in the annual mean. Stratocumulus clouds typically occupy the upper few hundred meters of the planetary boundary layer (PBL) (Wood, 2012). They can absorb atmospheric long-wave radiation and reflect solar short-wave radiation to influence the radiation budget of the Earth's atmospheric system (Pyrina et al., 2015; Ramanathan et al., 1989; Zelinka et al., 2014).

We supplement the introduction of Zn in lines 164-167.

Define the relative heights of the cloud as Zn:

$$Zn = \frac{Z - Z_{base}}{Z_{top} - Z_{base}}$$

In the formula, $Z_{base}$ is the height of the cloud base, and $Z_{top}$ is the height of the cloud top. The cloud heights have been normalized by setting the cloud base as 0 and the cloud top as 1.

References:

King M D, Platnick S, Menzel W P, et al. 2013. Spatial and temporal distribution of clouds observed by MODIS onboard the Terra and Aqua satellites [J]. IEEE transactions on geoscience and remote sensing, 51(7): 3826-3852. https://doi.org/10.1109/TGRS.2012.2227333

Wood R. 2012. Stratocumulus clouds [J]. Monthly Weather Review, 140(8): 2373-2423. https://doi.org/10.1029/2010GL043337 https://doi.org/10.1175/MWR-D-11-00121.1

Pyrina M, Hatzianastassiou N, Matsoukas C, et al. 2015. Cloud effects on the solar and thermal radiation budgets of the Mediterranean basin [J]. Atmospheric Research, 152: 14-28. https://doi.org/10.1016/j.atmosres.2013.11.009

Ramanathan V, Cess R, Harrison E, et al. 1989. Cloud-radiative forcing and climate: Results from the Earth Radiation Budget Experiment [J]. Science, 243(4887): 57-63. https://doi.org/10.1126/science.243.4887.57

Zelinka M D, Andrews T, Forster P M, et al. 2014. Quantifying components of aerosol-cloud-radiation interactions in climate models [J]. Journal of Geophysical Research: Atmospheres, 119(12): 7599-7615. https://doi.org/10.1002/2014JD021710

3. This study only includes 9 aircraft observations during daytime between 10:00 and 20:00 hours. Nighttime observations are completely missing. Thus, the authors could not assert that the results from this study can present diurnal variabilities of stratocumulus clouds in the region.

Response: Thank you for highlighting this limitation in our research. We have revised the title, subtitles and content related to "diurnal variabilities" in the manuscript. We acknowledge that the lack of nighttime observations limits our ability to capture regional diurnal variations in stratocumulus clouds adequately. Our study is based on observations made by nine aircraft during the day, when boundary layer

height changes significantly, and the vertical transport activity of aerosols is intense, leading to a more pronounced and rapid impact on clouds. We recognize that including nighttime data would provide a more complete picture of these clouds. Future studies should address this gap by combining daytime and nighttime observations. We have ensured that this limitation is clearly stated in the manuscript.

We have changed the title to "Aircraft observations of aerosol and microphysical quantities of stratocumulus in autumn over Guangxi Province, China: temporal evolution characteristics, vertical distribution, and aerosol-cloud interactions" and the title of Section 3.2 to "Vertical distribution and temporal variation of cloud microphysical quantities."

4. In section 2.1, the number concentration of aerosol particles corresponds to particles between 0.11 and 3.0 μm, and the number concentration of cloud droplets have sizes between 2.0 and 50 μm. Does the size refer to optical size or aerodynamic size? If they are the same size with the same definition, there is an overlap between the two size ranges. Would the counted aerosol particles include droplets between 2.0 and 3.0 μm? or would the counted droplets include aerosol particles between 2.0 and 3.0 μm? How could this influence the results in this study? This was not evaluated/addressed in this manuscript.

Response: Thank you for your comment. The size ranges for aerosol particles and cloud droplets refer to their optical sizes. The particle number concentration for the minimum size range (< 3 μm) measured by FCDP exhibits significant uncertainty. Consistent with previous studies (Wei et al., 2022; Wu et al., 2022), this study excludes cloud droplets smaller than 3.0 μm in Nc and cloud droplet size distribution, ensuring that there is no overlap between aerosol and cloud droplet number concentrations and size distributions. To provide a more precise understanding for the readers, we have added the corresponding cloud droplet size range for Nc in lines 133-135.

References:

Wei L, Huang M, Zhang R, et al. 2022. Microphysical characteristics of precipitating cumulus cloud based on airborne Ka-band cloud radar and droplet measurements[J]. Atmospheric

and Oceanic Science Letters, 15(2): 100134. https://doi.org/10.1016/j.aosl.2021.100134

Wu X, Wang M, Zhao D, et al. 2022. The microphysical characteristics of wintertime cold clouds in North China[J]. Advances in Atmospheric Sciences, 2022, 39(12): 2056-2070. https://doi.org/10.1007/s00376-022-1274-4

5. Are the number concentrations of aerosol particles and cloud droplets in volumetric unit or are they already corrected to the standard atmospheric condition? The presented results in Figure 2 are from different days and different altitudes, they should be corrected to the standard atmospheric condition before being combined in one figure.

Response: Thank you for your comment. Aerosol and cloud droplet concentrations are measured in volume units. The PCASP-100X and FCDP probes record the sampling flow rate, and the sampling volume is determined by multiplying the flow rate by the sampling time. The software then calculates the number concentration and particle size distribution based on the actual sampling volume. The data in this study are processed using software provided by the supplier to ensure accurate concentrations of aerosols and cloud droplets and their distribution spectrum. This study's data quality control method is the same as that of previous studies (Zhang et al., 2023; Zhao et al., 2023), and quality control has been completed. According to our previous research (Liu et al., 2024), there were no special weather processes in the upper air and on the ground in Guangxi Province during the observation period, which ensured the quality of the data and the universality of the conclusions.

References:

Zhang D, Vogelmann A M, Yang F, et al. 2023. Evaluation of four ground-based retrievals of cloud droplet number concentration in marine stratocumulus with aircraft in situ measurements[J]. Atmospheric Measurement Techniques, 16(23): 5827-5846. https://doi.org/10.5194/amt-16-5827-2023

Zhao D, Wang M, Rosenfeld D, et al. Aircraft observation of fast initiation of mixed phase precipitation in convective cloud over the Tibetan Plateau[J]. Atmospheric Research, 2023, 285: 106627. https://doi.org/10.1016/j.atmosres.2023.106627

Liu S, Wang H, Zhao D, et al. 2024. Aircraft observations of aerosols and BC in autumn over

Guangxi Province, China: Diurnal variation, vertical distribution and source appointment [J]. Science of The Total Environment, 906: 167550. https://doi.org/10.1016/j.scitotenv.2023.167550

6. The authors stated the source apportionment of airmasses and tried to use the property difference between continental and marine aerosols to explain the differences in observed cloud properties. However, how was the source apportionment performed in this study? Or was it published in parallel studies? What are the physiochemical properties of these two kinds of air masses? like size distribution and hygroscopicity which are important for acting as cloud condensation nuclei.

Response: Thank you for your comment. The sources of aerosols were analyzed using the Hybrid Single-Particle Lagrangian Integrated Trajectory (HYSPLIT) by the National Oceanic and Atmospheric Administration. We found that below 3000 m, air masses that have passed over land tend to bring higher aerosol number concentrations to Guangxi compared to those that have passed over the ocean. This provides sufficient condensation nuclei for cloud formation. Additionally, the aerosol size distribution from land is broader than that of marine aerosols, and aerosols with a diameter greater than 1 μm are more easily influenced by continental air masses (Liu et al., 2024). Larger diameter cloud condensation nuclei are also generally more likely to activate than smaller nuclei. Therefore, in this study, we categorized and discussed the observational data based on air mass sources and found that under the influence of two air mass sources, LWC in this area are similar, further demonstrating that aerosol number concentration is an essential factor affecting the microphysical properties of clouds in this region.

Reference:

Liu S, Wang H, Zhao D, et al. 2024. Aircraft observations of aerosols and BC in autumn over Guangxi Province, China: Diurnal variation, vertical distribution and source appointment [J]. Science of The Total Environment, 906: 167550. https://doi.org/10.1016/j.scitotenv.2023.167550

7. It is well known that aerosol particle number concentration and updraft velocity (i.e., water supersaturation conditions) are the two most important factors for the

observed cloud droplet number concentrations. Although the authors presented the latitudinal profiles of vertical pressure velocity at different times (Figure 5), the discussions on the single effect of updraft velocity and the combination effects of these two factors are still lacking. The results in Figure 5a at 08:00 are irrelevant to results in Figures 2, 3 and 4, whereas the required results at 10:00, 12:00, 13:00, 15:00, 16:00 and 18:00 are missing. Also, the x-axis in Figure 5 is not labelled, it is neither indicated in the caption nor clarified in the text. The y-axis might have an unit in Pascal.

Response: Thank you for your comment. In vertical detection during aircraft observations, the vertical wind speed and direction measurements are often inaccurate. Therefore, reanalysis data illustrates the state of vertical atmospheric motion (Ge et al., 2021; Muhlbauer et al., 2014). The MERRA2 reanalysis data used in this study has a vertical pressure velocity resolution of 3 hours, which results in missing data for the times 10:00, 12:00, 13:00, 15:00, 16:00, and 18:00. We have described the characteristics for these specific times and supplemented the discussion on the joint effects of aerosol particle number concentration and updraft velocity on Nc. Additionally, we have revised the x-axis labels in Fig 5 and changed the y-axis to height for easier comparative analysis with Figs 2, 3, and 4.

References:

Ge J, Wang Z, Wang C, et al. 2021. Diurnal variations of global clouds observed from the CATS spaceborne lidar and their links to large-scale meteorological factors [J]. Climate Dynamics, 57: 2637-2651. https://doi.org/10.1007/s00382-021-05829-2

Muhlbauer A, Kalesse H, Kollias P. 2014. Vertical velocities and turbulence in midlatitude anvil cirrus: A comparison between in situ aircraft measurements and ground-based Doppler cloud radar retrievals[J]. Geophysical Research Letters, 41(22): 7814-7821. https://doi.org/10.1002/2014GL062279

Specific comments:

Title: better to have 'aerosol-cloud interactions' rather than 'aerosol-cloud relationship'

Response: Thank you for your comment. We have changed the title to "Measurement report: Aircraft observations of aerosol and microphysical quantities of stratocumulus in autumn over Guangxi Province, China: temporal variation, vertical distribution and aerosol-cloud interactions."

Line 24: 'boundary layer (PBL)' or 'planetary boundary layer (PBL)'?

Response: We have changed it to "planetary boundary layer."

Line 25-26: I am confused about the sentence 'The lower layer of the stratocumulus cloud in Guangxi was mainly small particle-size cloud droplet' and also the phrase 'small particle-size cloud droplet'. How could the layer be cloud droplets? Did you intend to state that the layer mainly contain small-sized cloud droplets? Or did you intend to state that the layer mainly contain cloud droplets which are formed by small-sized aerosol particles (i.e. small-sized cloud condensation nuclei)?

Response: Thank you for your comment. We have carefully reviewed and revised all unclear statements in the manuscript to improve clarity and ensure that the content is accurately conveyed. In lines 25-26, we have changed the statement to "The lower layer of the stratocumulus cloud in Guangxi mainly contained small-sized cloud droplets."

Line 28: Does "cloud microphysical quantity" refer to Nc and Ed?

Response: Thank you for your comment. In our manuscript, the term "cloud microphysical quantity" indeed refers to both the cloud droplet number concentration (Nc) and the cloud droplet effective diameter ($E_d$).

Line 29: Didn't you define Nc already in Line 22?

Response: Thank you for your comment. We have reviewed the manuscript and ensured that the abbreviation is clearly defined at its first appearance and only once. We have removed the redundant definition in line 29 to improve the flow and clarity of the text.

Line 32: Did you mean the lower layer of the PBL will show increases in small-sized cloud droplets? It would be good to increase the clarity of the statement.

Response: Thank you for your comment. In line 32, we have changed the statement to "Aerosols from the free atmosphere were transported into PBL (10:00 to 13:00), providing an abundance of cloud condensation nuclei, which led to an increase in the number of small particle-size cloud droplets in the lower layer of the cloud (near the top of PBL)".

Line 34: Na was defined in Line 22 already.

Response: Thank you for your comment. We removed the redundant definition of Na in line 35.

Line 39: What does 'FIE' mean?

Response: Thank you for your comment. We have added the definition of FIE in line 40. $E_d$ and Na remain negatively correlated in different liquid water content ranges, and FIE (the aerosol first indirect effect) ranged from 0.07 to 0.58.

Line 40: It should be 'The boundary layer' or' The planetary boundary layer'

Response: Thank you for your comment. As suggested, we have changed from " The boundary layer " to " The planetary boundary layer " in line 47.

Line 44-45: Stratocumulus clouds cannot account for land or water surfaces but can account for clouds that are over land or water surfaces.

Response: Thank you for your comment. We have changed the statement to "Clouds are an essential component of the Earth-atmosphere system, covering over 67 % of the Earth's surface (King et al., 2013), with stratocumulus clouds covering approximately 20 % of Earth's surface in the annual mean" in lines 44-46.

Line 56-57: Please correct the grammar in the statement "… an increase of aerosol number concentration (Na) would increase in cloud droplet number concentration (Nc) and a decrease in cloud droplet size, …"

Response: Thank you for your comment. We have corrected the grammatical errors in the statement and revised the sentence to "Twomey (1977) suggested that, with the liquid water path of clouds remaining constant, an increase in aerosol number concentration (Na) would lead to an increase in cloud droplet number concentration (Nc) and a decrease in cloud droplet size, thereby enhancing cloud albedo" in lines 58-61.

Line 61: Change "ground-based remote" to "ground-based remote sensing"

Response: Thank you for your comment. As suggested, we have changed from "ground-based remote" to "ground-based remote sensing" in line 64.

Line 62: Change "between aerosol and cloud" to "between aerosols and clouds"

Response: Thank you for your comment. As suggested, we have changed from " between aerosol and cloud " to " between aerosols and clouds " in line 65.

Line 65: What is 'AOD'?

Response: Thank you for your comment. We have removed the abbreviation for aerosol optical depth to ensure clarity in the manuscript. We have changed the statement to "Kim et al. (2003) found that the aerosol optical depth in Oklahoma presents a linear proportional relationship with liquid water path (LWP) on a completely cloudy day with single-layer cloud, and the effective radius of cloud droplets is negatively correlated with the surface aerosol light scattering coefficient" in lines 68-71.

Line 67: should be 'cloud'

Response: Thank you for your comment. We have changed from "single-layer clouds" to "single-layer cloud" in line 70.

Line 70-72: In the sentence "Under high aerosol loading, smaller cloud droplets with higher droplet number concentration were observed under high aerosol loading, while fewer larger cloud droplets were formed under low aerosol loading.", how did you define low and high aerosol loadings? How do you compare higher and fewer cloud droplets? Numbers should be provided from the literature.

Response: Thank you for your comment. We have added data to enhance the rigor of the argument. We have changed the statement to "Under high aerosol loading (Na below the cloud base was 4573 $cm^{-3}$), smaller cloud droplets with higher droplet number concentration (Nc = 157 $cm^{-3}$) were observed, while fewer larger cloud droplets (Nc = 118 $cm^{-3}$) were formed under low aerosol loading (Na below the cloud base was 982 $cm^{-3}$)" in lines 74-76.

Line 72: Didn't you already define 'Cloud droplet number concentration' as 'Nc' in Line 57? And in Line 73, did you mean liquid water path (LWP defined in Line 65) by saying 'liquid water content'. It is not far away. The same problem was shown in the abstract. More carefulness in writing will be appreciated. No further comments will be provided on such abbreviations, including but not limited to Nc, Nd, Ed, LWP, and PBL. Please check through the whole manuscript. Detailed writing tips/rules can be found in ACP guidelines (via https://www.atmospheric-chemistry-and-physics.net/submission.html).

Response: Thank you for your comment. We have thoroughly reviewed the manuscript and corrected the use of abbreviations to avoid repeated definitions and redundant terms.

Line 74: Is it between aerosol number concentrations and the effective diameter of cloud droplets (Ed)? Or is it between one of any other aerosol properties and the Ed?

Response: Thank you for your comment. We have changed the statement to "However, some scholars have also observed a positive correlation between Na and the effective diameter of cloud droplets ($E_d$) (Harikishan et al., 2016; Jose et al., 2020; Liu et al., 2020), referred to as the anti-Twomey effect" in lines 78-80.

Line 77: Please refer to an exact study that directly compared aircraft observations with other methodologies and successfully demonstrated that aircraft observations are the most reliable method to investigate the vertical relationships between aerosols and clouds.

Response: Thank you for your comment. McFarquhar et al. (2021) provided an overview of the Southern Ocean Cloud Radiation and Aerosol Transport Experimental Study (SOCRATES), which focuses on clouds, aerosols, and precipitation. SOCRATES provided in situ observations critical for process studies and evaluating remote sensing retrievals. These observations were the only direct measurements of aerosols below, inside, and above clouds. The study compared these observations with remote sensing data and model simulations, demonstrating the capability of aircraft measurements to provide detailed vertical profiles of aerosol and cloud interactions. The authors emphasized the importance of aircraft observations in capturing the vertical distribution and microphysical properties of aerosols and clouds, which are often challenging to measure accurately using other methods.

Reference:

McFarquhar, G. M., and Coauthors, 2021: Observations of Clouds, Aerosols, Precipitation, and Surface Radiation over the Southern Ocean: An Overview of CAPRICORN, MARCUS, MICRE, and SOCRATES. Bull. Amer. Meteor. Soc., 102, E894 – E928, https://doi.org/10.1175/BAMS-D-20-0132.1.

Line 81-82: Please correct the grammar in 'This leads to cloud condensation nuclei

(CCN) variations and droplet concentration near cloud tops.'

Response: Thank you for your comment. We have changed the statement to "This leads to variations in CCN and Nc near cloud tops" in lines 85-86.

Line 87: It should be 'they showed'.

Response: Thank you for your comment. We have corrected the tense error and updated the relevant section to "they showed" in line 91.

Line 89: What is 'LWC'?

Response: Thank you for your comment. We have changed the statement to "It also shows that the relationship between the effective radius of cloud droplets and Na changes from negative to positive when liquid water content (LWC) increases" in lines 91-93 to ensure that the explanation of LWC is included.

Line 92: What are 'hair drops'?

Response: Thank you for your comment. We have changed the statement to "Lu et al. (2007) compared the microphysical quantities of stratocumulus clouds influenced by aircraft flight tracks and those in undisturbed regions and found that the effective radius of cloud droplets in the flight path region was smaller, the number concentration of cloud drops was lower, and the cloud LWC was larger, providing observational evidence for the first indirect effect of aerosols" in lines 93-97.

Line 93: The first indirect effect of what? Or it refers to Twomey effect? It is not clear and the readability should be improved.

Response: Thank you for your comment. We have supplemented the section in the manuscript where the Twomey effect is first mentioned in line 67 and ensured that subsequent descriptions of the Twomey effect and the first indirect effect of aerosols are clear to improve readability.

Line 106-108: Is the statement relevant for your findings or are those findings are from Liu et al. 2024? "

Response: Thank you for your comment. These are the results of our previous research, which demonstrated the diurnal variation characteristics of aerosol vertical transport. This has further sparked our interest in the interactions between aerosols and clouds. We hope to explain the diurnal variations of cloud microphysical

properties from the perspective of the diurnal variation of aerosol vertical distribution.

Line 108-109: The statement of 'aerosols can have microphysical properties of clouds' sounds very wired.

Response: Thank you for your comment. We have changed the statement to "Previous studies have shown that aerosols can affect cloud microphysical properties. When aerosol particles settle onto clouds or the cloud top is elevated, aerosols can alter the microphysical characteristics of clouds by being entrained into the cloud top" in lines 109-112.

Line 114-115: 'We demonstrated that this region's correlation between aerosols and clouds conforms to the Twomey effect.' The structure and grammar of this sentence can be much better developed.

Response: Thank you for your comment. We have changed the statement to "Our findings indicate that the interaction between aerosols and clouds in this region aligns with the Twomey effect" in lines 116-118.

Line 126: Is it 'accuracy' or 'uncertainty'?

Response: Thank you for your comment. We have changed the statement to "A passive cavity aerosol spectrometer probe (PCASP-100X, DMT Inc, USA) was installed to provide aerosol number concentrations in the particle size range of 0.11 to 3 μm, with a time resolution of 1s, particle size uncertainty of 20 %, and concentration uncertainty of 16 %" in lines 125-128.

Line 134-137: Average values should be provided.

Response: Thank you for your comment. We have added an explanation for the data in Table 1. The values of Na, Nc, LWC, and $E_d$ in the table are all averages, with their standard deviations.

Line 139-140: This statement contradicts the other statements. How do you mean by saying 'close to'? Does it mean 'similar to'? More rigorous and clearer logic to develop statements would be appreciated.

Response: Thank you for your comment. We have revised such statements to ensure accurate wording. We have changed the statement to "The cloud microphysical characteristics of stratocumulus observed in this study are similar to those of previous

observations."

Line 140-141: Why to have capitalized 'Stratocumulus'?

Response: Thank you for your comment. We have changed "Stratocumulus" to "stratocumulus" to ensure grammatical correctness.

Line 150: The observations from 08:00 to 20:00 hours do not represent diurnal observations. Also, relevant statements/conclusions on diurnal variations do not make sense based on daytime observations only.

Response: Thank you for your comment. We have revised the title, subtitles and content related to "diurnal variabilities" in the manuscript.

Line 155: The LWC$\geq 10^{-3}$ g·m$^{-3}$ is ambiguous.

Response: Thank you for your comment. We explained liquid water content (LWC) in line 54. The commonly used standards for determining conditions within warm clouds are Nc $\geq 10$ cm$^{-3}$ and LWC $\geq 10^{-3}$ g·m$^{-3}$.

Line 158: Is the observation height above seal level or ground level?

Response: Thank you for your comment. The observation height is above sea level.

Line 161-163: What are ni, ri, and ρw?

Response: Thank you for your comment. We have added explanations for the relevant parameters in the formula to ensure clarity in the presentation. In the formulas, $n_i$ is the cloud number concentration for each bin. $r_i$ is the median particle size for each bin. $\rho_w$ is the density of water.

Line 166: A full stop is missing.

Response: Thank you for your comment. We have added a full stop in line 167.

Line 170: 'Where is the amount of α aerosol', this statement does not make sense.

Response: Thank you for your comment. We have changed the statement to "α represents the physical quantity of aerosols, which can be quantified using aerosol optical depth (Feingold et al., 2001), aerosol extinction coefficient (Feingold et al., 2003), cloud condensation nuclei concentration, and aerosol number concentration (Che et al., 2021; Zhao et al., 2012; Zhao et al., 2018). The FIE value may vary with the variables representing the aerosol amount." in lines 171-175.

Line 176-179: Sentence too long and wrong grammar. Also, how cloud the same Na represent two parameters of two cases (interstitial aerosols and out-cloud aerosols)? How did you calculate the average values? Is it calculated at the same height of nine flights? Or is it calculated as the average of observations in the height interval (10 m) for each flight and then you present the results if nine flights together? This should be clearly introduced in the manuscript.

Response: Thank you for your comment. We have reformulated the statement and added the calculation method of the data in lines 187-193. We distinguished the aerosol, cloud droplet, and meteorological data inside and outside the cloud based on the criteria of $Nc \geq 10$ cm$^{-3}$ and LWC $\geq 10^{-3}$ g·m$^{-3}$. The vertical averages were calculated at 10 m height intervals, resulting in the vertical distribution of physical quantities from nine observational flights, covering a height range of 0-4000 m, ensuring that the vertical resolution of each physical quantity was consistent. We then calculated the mean vertical distribution of the physical quantities from the nine observational flights, resulting in the vertical distribution maps of each physical quantity during the observation period, as shown in Fig.1. Fig.1.a illustrates the vertical distribution of Nc and Na, where aerosols within the cloud are referred to as interstitial aerosols, while aerosols outside the cloud are termed out-cloud aerosols. Fig.1b presents the vertical distribution of LWC and $E_d$, calculated based on the cloud droplet size distribution and Nc. Fig.1c and 1d show the vertical distribution of atmospheric temperature and relative humidity inside and outside the cloud, respectively. This figure represents the vertical distribution of cloud droplets, aerosols, and meteorological elements throughout the entire observation period, reflecting the environmental characteristics of stratocumulus clouds in Guangxi during this period.

Line 180-181: It is very hard to read these two average values in Figure 1a.

Response: Thank you for your comment. We have adjusted the Fig.1a to ensure that the data is clear.

Line 181-185: The authors contradict themselves. They first stated that 'Nc decreased first and then remained unchanged with the increase of height.' On the contrary, they stated that 'Between 1500 and 3300 m, the Nc changed little with the increase of

height, and the range of Nc is 10-200 cm$^{-3}$ (Fig. 1a).'

Response: Thank you for your comment. We have adjusted the expression of the vertical distribution characteristics of Nc in lines 197-204.

Below 1500 m, Nc first decreases and then stabilizes with increasing height, following a trend similar to Na's. This indicates that the number of cloud condensation nuclei capable of activating cloud droplets diminishes as altitude increases. Compared to the upper atmosphere (above 1500 m), there are more cloud condensation nuclei in the lower atmosphere, resulting in an average Nc value of 407 cm$^{-3}$. Between 1500 m and 3300 m, Nc shows little variation with height, remaining concentrated around 100 cm$^{-3}$ at each altitude (Fig.1a). The lower Nc observed at certain altitudes may be due to the observation area being close to the edge of the cloud.

Line 190: Altitude values for each stage should be specified.

Response: Thank you for your comment. We have increased the heights of the PBL in Figs 2, 3, and 4 to ensure that the reader can clearly see the content and better understand the relevant information.

Line 192-193: The reason why there exists an inversion layer for temperatures should be shortly but clearly explained. Why such a temperature inversion layer hamper cloud droplet growth should be elaborated.

Response: Thank you for your comment. We have added information regarding the reasons for the presence of temperature inversion layers and the mechanisms that hinder cloud droplet growth in lines 211-214.

In Guangxi, the top of PBL during autumn ranges from 1000 to 1500 m, where temperature inversion layers occur. This temperature structure increases the stability of the air, suppressing the formation of vertical airflow and hindering the growth of cloud droplets.

Line 197: The results presented in this study cannot represent diurnal variations since the night results are missing.

Response: Thank you for your comment. We have revised the title, subtitles and content related to "diurnal variabilities" in the manuscript.

Line 202-205: Sentence too long and with a poor structure.

Response: Thank you for your comment. We have adjusted to shorten the sentence length and improve the structure to enhance readability. We have changed the statement to "To understand the time variation of the vertical distribution of cloud microphysical quantities, the data were classified. Vertical profiles of interstitial aerosol (Na), Na outside the cloud (Fig. 2), cloud microphysical quantities (Fig. 3), and meteorological elements inside and outside the cloud (Fig. 4) were obtained at ten times from 10:00 to 18:00 and at 20:00. The data collected inside the cloud were original, while the average values outside the cloud were calculated at 10 m intervals" in lines 224-228.

Line 215: Is Nc less than 100 cm-3 or is it lower than the detection limit of the instrument? What is the detection limit of the instrument? There are only a few data point for altitudes below 900 m.

Response: Thank you for your comment. The FCDP probe is an instrument mounted under the wing of an aircraft that detects the size of a laser beam scattered when a single droplet passes through it. Consequently, the detection limit of the FCDP is $1 \text{ cm}^{-3}$. Under typical conditions, we consider Nc $>10 \text{ cm}^{-3}$ to indicate the presence of a cloud. Table 1 presents nine flights' observed stratocumulus cloud top and cloud base heights. Only two flights recorded stratocumulus cloud base heights below 900 m, resulting in a limited amount of Nc data below this altitude. Additionally, there were instances (such as at 11:00 and 15:00) when the minimum altitude of the aircraft during vertical detection exceeded 900 m, leading to a scarcity of Na, temperature, and relative humidity data points below 900 m.

Line 217: Figure 4a was mentioned earlier than Figure 3. So, why not switch these two figures.

Response: Thank you for your comment. We have adjusted the figures' order to ensure the content's logical flow.

Line 217: Why only temperature only is the reason? Is RH irrelevant? are the aerosol size and other physiochemical properties irrelevant?

Response: Thank you for your comment. We have revised the description of the vertical distribution characteristics of cloud microphysical properties at 10:00. At

10:00, Nc below 900 m was less than 100 cm$^{-3}$, and Na in PBL was high (Fig. 2a). Although there were sufficient aerosols that can be activated into cloud condensation nuclei, RH > 60 %, the atmospheric temperature was high, which was not conducive to the activation of small-size aerosol particles (Fig. 3a) in lines 236-240.

Line 222: Which part of the results shows that water vapor is sufficient?

Response: Thank you for your comment. We have added the average value of LWC in line 243. Above 1500 m, although the water vapor condition was sufficient (LWC =0.16 g·m$^{-3}$), the cloud condensation nucleus was few, resulting in an average Nc value of only 35 cm$^{-3}$.

Line 225-228: Poor sentence structure. Is there any statistical results or results from Nc that could support this statementstating that the temperature inversion layer and the increase in Na are correlated? Between 1500-1600 m, there are approximately 10 data points only.

Response: Thank you for your comment. We have revised the wording of the sentences in lines 247-254. At 1500 m, Na (interstitial aerosol) was 34 cm$^{-3}$, increasing to 134 cm$^{-3}$ at 1600 m. RH remained nearly constant in this range, while LWC rose from 0.16 to 0.19 g·m$^{-3}$, promoting the hygroscopic growth of aerosols. However, Nc did not show a significant increase. Thus, the inversion layer within the cloud may contribute to the rise in Na (interstitial aerosol). This increase suggests more aerosols are inactive or unable to activate within the cloud. These aerosols may result from mixing warm air from outside the cloud at the cloud base. Furthermore, the inversion layer may hinder vertical airflow within the cloud, suppressing cloud droplet growth.

Line 229: Where is the top of PBL? How was the PBL height determined/calculated?

Response: Thank you for your comment. We have added the heights of PBL to Figs 2, 3, 4, and 9 and supplemented the calculation method for PBL height in lines 154-156.

Line 229-232: Very poor sentence structure.

Response: Thank you for your comment. We have carefully reviewed and improved the relevant sections to ensure more precise and fluid expression. At 11:00,

aerosols were transported by updrafts (Fig. 5) to around 1500 m (near the top of PBL) and activated into cloud condensation nuclei. Below 1500 m, the average Nc value was 102 cm⁻³, while the average LWC value was only 0.03 g·m⁻³. Cloud droplets were competing for water vapor, and the Ed value was only 8.20 μm, which was similar to the cloud microphysical characteristics near the PBL at 10:00. Between 1500 and 3150 m, Na was less than 10 cm⁻³, indicating insufficient cloud condensation nuclei and the average Nc was only 29 cm⁻³. Compared to 10:00, the LWC was higher (mean 0.19 g·m⁻³), resulting in a larger Ed in the upper part of the cloud, with an average of 28.95 μm.

Line 240-242: Can the pressure be calculated into altitude values? This way, it will be easier to compare to the results in Figures 2, 3 and 4.

Response: Thank you for your comment. We have changed the y-axis of Fig 5 to elevation to facilitate comparison and analysis with Figs 2, 3, and 4.

[Figure]

**Fig 5** Latitudinal profiles of vertical pressure velocity at different times in Guangxi (solid black line is the latitude range observed by aircraft, and a positive value is downdraft, a negative value is updraft, a is 11:00, b is 14:00, c is 17:00, d is 20:00)

Line 246: What do 'different locations' mean? Air mass regions? Or vertical locations?

Response: Thank you for your comment. We have revised the wording of the sentences. At 13:00, the Nc ranged from 13 to 2052 cm⁻³ below 1200 m, which may be attributed to the uneven development of clouds within the detection range.

Line 247: Figure 5b is for 11:00 and Figure 5c is for 14:00. How could these two panels be used for discussions on 13:00?

Response: Thank you for your comment. We have revised the wording of the sentences. The increase in solar radiation leads to higher near-surface temperatures (T > 25 °C), which enhances turbulent activity within the PBL and is favorable for cloud droplet formation. Therefore, Nc at 13:00 is more significant than that at 10:00, and many cloud droplets hinder their particle size growth, with an average $E_d$ value of 9.23 µm.

Line 248: what does 'more significant' mean? Is there a difference between those two beyond the uncertainty? It is also very ambiguous too say 'many cloud droplets', how many? From where one can read the quantity of the cloud droplets mentioned by the authors? Line 250: A strong inversion layer of what?

Response: Thank you for your comment. We have carefully reviewed and improved the relevant sections to ensure more precise and fluid expression. From 1200 m to 1500 m, the mean values of Nc and $E_d$ were 155 cm$^{-3}$ and 12.29 µm. At this height, a strong inversion layer appeared, and cloud droplet evaporation activity was enhanced (Li et al., 2003), resulting in a higher Na (interstitial aerosol) than Na (out cloud). For Stratocumulus clouds above 1500 m, the Nc varied little with height, and the average $E_d$ was 21.45 µm.

Line 254: What does 'the top height of PBL was the highest' mean? Can the height of the PBL be compared to Nc number concentrations? From where, it presents the results of PBL height? Line 257: An inversion layer of what?

Response: Thank you for your comment. We have carefully reviewed and improved the relevant sections to ensure more precise and fluid expression. At 14:00, the Nc range below 1500 m was 11 to 1109 cm$^{-3}$, with the highest PBL top height at 1500 m, which diluted the Na (out of the cloud) within the PBL, resulting in a decrease in the maximum Nc (Nc = 1109 cm$^{-3}$). The average LWC was 0.29 g·m$^{-3}$ (Fig. 3e), higher than at 13:00, providing moisture conditions for cloud droplet growth, while the upward airflow was strong (Fig. 5b). Consequently, the average Ed was 13.75 µm. An inversion layer was present at 2500 m, hindering aerosol diffusion and

enhancing the evaporation of cloud droplets near the cloud top, leading to a peak in Na (interstitial aerosol) at that height.

Line 260: more cloud droplets than what?

Response: Thank you for your comment. We have carefully reviewed and improved the relevant sections to ensure they are clear. At 15:00, the Nc and Na (interstitial aerosol) between 1600 m and 2000 m were higher than those at 14:00, with average values of 720 cm$^{-3}$ and 249 cm$^{-3}$ (Fig. 2f).

Line 262-263: There are no results in Figure 5 or other figures to show the updraft. This interpretation is not groundless. Line 263-264: Based on results from which Figure could the authors draw such a statement? Please specify.

Response: Thank you for your comment. We have revised the wording of the sentences in lines 292-296. Although the moisture conditions were sufficient, with an average LWC of 0.23 g·m$^{-3}$, which was higher than the 0.05 g·m$^{-3}$ recorded at 14:00 (Fig. 3f), and RH was 52% (Fig. 2f), the average $E_d$ decreased to 16.73 μm (Fig. 3f). This decrease was due to the competition for moisture among cloud droplets, which led to an increase in small particle-size cloud droplets.

Line 268: How do authors define low temperatures and high humidities? The development of the statement should be clarified.

Response: Thank you for your comment. We have revised the wording of the sentences in lines 301-303. The low temperature (T = 7.75 °C) and high humidity (RH = 70 %) of the cloud environment (Fig.4) were conducive to the activation of aerosol. The maximum value of Nc reached 395 cm$^{-3}$. However, the average of $E_d$ was only 17.13 μm due to water vapor contention between cloud droplets.

Line 271: How could it be known that 'At 17:00, the height of the top of PBL decreased, and part of the aerosol remained above PBL,….'? Line 273: What does 'Ed concentrated in the 10-20 μm' mean? given that Ed is defined as a size.

Response: Thank you for your comment. We have added supplementary information in lines 305-310. At 17:00, the height of PBL decreased to 730 m. Aerosols were transported above the PBL (Fig. 2h), providing CCN above 2000 m.

Nc remained constant with an average of 134 cm$^{-3}$ (Fig. 2h), while E$_d$ averaged 17.12 μm (Fig. 3h). Under the cooling of the atmosphere and the cooling of the cloud tops at sunset, the E$_d$ near the cloud tops is greater than 30 μm. The temperature inversion layer of 1600-2000 m (Fig. 4h) enhanced cloud droplet growth and hindered aerosol diffusion, causing the Na (interstitial aerosol) to be higher than the Na out cloud.

Line 277: The same comment on the statement '… resulting in Na (interstitial aerosol) being more significant than Na (out cloud)' as the comment on Line 248.

Response: Thank you for your comment. We have carefully reviewed and improved the relevant sections to ensure clearer and more fluid expression. We have changed the statement to "The temperature inversion layer at altitudes of 1600 to 2000 meters (Fig. 4h) facilitated the growth of cloud droplets while hindering the diffusion of aerosols. This led to the size of Na (interstitial aerosol) being greater than that of Na (out of cloud)" in lines 308-310.

Line 299: 'Na and Nc in the cloud lower layer were large', how large they are? And the results refer to which figure and which panel? Line 299-300: It is very awkward to have 'Na and Nc in the cloud lower layer were large' and 'large cloud droplet was less'. Line 301-302: 'Na and Nc the upper cloud layer are the smallest', should be in the upper cloud layer. Line 302: How do the authors know the water vapor is sufficient?

Response: Thank you for your comment. We have revised the wording in this section. Below is our modified content.

From 10:00 to 13:00, the interstitial aerosol particle size in the cloud's lower layer was concentrated below 0.4 μm. In comparison, the cloud droplet diameter was primarily concentrated below 20 μm, with few large particle-size cloud droplets (Fig. 6a, 6d). In the middle cloud layer, Na across all particle size ranges had decreased to below 1000 cm$^{-3}$·μm$^{-1}$. Nc for particles smaller than 20 μm has decreased, while Nc for particles larger than 20 μm exceeded 0.1 cm$^{-3}$·μm$^{-1}$(Fig. 6b, 6e). Na in the upper cloud layer was minimal compared to the middle and lower layers. Sufficient water vapor (LWC = 0.14 g·m$^{-3}$, Fig.3a-c) and low temperature (T = 11.72 °C, Fig.4a-c) promote the growth of cloud droplets, resulting in fewer Nc for particles larger than 20 μm in the upper layer (Fig. 6c, 6f) compared to the middle layer.

Line 305: 'the updraft diffused aerosols in the lower atmospheric layer upward' read awkward. Better wording is needed. Line 305-307: 'From 14:00 to 16:00, the updraft diffused aerosols in the lower atmospheric layer upward, and Na and Nc in the lower cloud layer decreased, which was conducive to the condensation growth of cloud droplets, and the number of cloud droplets with a diameter greater than 20μm increased.' One cannot understand what is the cause and what is the effect in the statement. This sentence is also very poorly organized.

Response: Thank you for your comment. We have changed the statement to "From 14:00 to 16:00, the upward airflow promoted the vertical diffusion of lower layer aerosols, resulting in a decrease in the Na and Nc in the cloud's lower layer. This change facilitated the condensation growth of cloud droplets, thereby increasing the number of cloud droplets with diameters exceeding 20 μm" in lines 338-343.

Line 312: Which part of the results in this manuscript shows the downdraft of the PBL between 17:00 and 20:00?

Response: Thank you for your comment. We have added the heights of PBL to Figs 2, 3, 4, and 9.

Line 316-317: What is the minor reasons? Based on which kind of comparisons, can the authors be sure the major reason?

Response: Thank you for your comment. We have changed the statement to "During this period, Nc was higher than that observed from 10:00 to 13:00. The increase in Nc may be attributed to the rise in Nc of droplets smaller than 20 μm" in lines 346-347.

Line 318: What are stratified clouds? Are they the same as or different from the aforementioned stratocumulus? No further information can be found in the following text.

Response: Thank you for your comment. Considering the feedback from other reviewers, we have changed the statement to "Verification of the Twomey Effect."

Line 319: Which previous studies? Any references? Are the airmass source apportionment in those studies relevant for the same period of observations in this study? Line 320: What does it mean by stating that 'air masses from land will bring

higher aerosols'? Higher aerosol particle number concentrations? Or aerosols in higher altitudes?

Response: Thank you for your comment. We have supplemented the relevant literature citations to ensure that the air mass source distribution in the cited studies corresponds to the same period as the observations in this research.

We have changed the statement to "Previous studies have shown two sources of aerosols in Guangxi, namely the land and the ocean, where air masses from land will bring higher aerosol particle number concentrations (Liu et al., 2024)" in lines 349-351.

Line 319-337: Without a clear definition for y-axis in Figure 7, the corresponding discussions do not make sense.

Response: Thank you for your comment. The y-axis of Fig.7 represents the number of samples. We have defined the y-axis in the figure caption of Fig.7.

Line 338-369: It is unbelievable to discuss the results in Figure 8 without a unit for the colormap.

Response: Thank you for your comment. We have supplemented the units of the color column to ensure that the data is clearly readable.

Line 365-367: 'The aerosol spectral patterns were similar', similar to what? It is nonsense to do aerosol source apportionment in such a baseless way. Did the authors analyse the physiochemical properties of those aerosols samples and compare the results to those of aerosol pollution sampled in surrounding areas of Guangxi? Or did the authors conduct simulation experiments (e.g. HYSPLIT or FLEXPART) to calculate the trajectory of these airmasses? Even if the source apportionment is based on previously relevant studies, at least, the relevant publications should be cited. Line 367-368: Again how can one read the PBL height in Figure 8?

Response: Thank you for your comment. We have rephrased the vertical distribution characteristics of cloud microphysical properties observed on October 29 and November 2, as well as the impact of aerosols on these cloud microphysical properties in lines 377-388 and 394-402.

On October 29, the aerosol pollution in PBL was severe (Fig.9a, $Na = 1331$ cm$^{-3}$).

The aerosol number concentration spectrum exhibited a bimodal distribution, with peak diameters of 0.14 and 0.22 μm (Fig.8a). The atmosphere contained sufficient CCN, resulting in a large Nc (Fig.9a, Nc = 460 cm$^{-3}$). As shown in the cloud droplet number concentration spectrum (Fig. 8c), most cloud droplets were concentrated in the size range of 3-24 μm (Fig.8c). $E_d$ was 9.69 μm (Fig. 9e), primarily because many cloud droplets competed for water vapor, making it difficult for them to grow into larger droplets. A strong inversion layer at 1500 m (Fig.9c) hindered the upward transport of aerosols. Consequently, Na above 1500 m was low, leading to a reduced Nc, with an average of only 35 cm$^{-3}$. Fig. 8c showed that cloud droplet sizes within the PBL primarily range from 8 to 21 μm. In contrast, above the PBL, cloud droplet sizes are mainly distributed below 8 μm and above 21 μm, with an average effective diameter ($E_d$) of 25.28 μm (Fig. 9e). These large particle-size cloud droplets likely originated from the collision and growth of droplets within the 8.0 to 21 μm range.

On November 2, Na in PBL (Fig. 9b, Na = 405 cm$^{-3}$) was slightly higher than Na in the upper air (Na = 220 cm$^{-3}$). The Nc in PBL (Nc = 243 cm$^{-3}$) was higher than that above PBL (Nc = 124 cm$^{-3}$). The concentration spectra of cloud droplet numbers exhibited a bimodal distribution (Fig. 8d). The presence of a large number of small cloud droplets in the PBL hinders the growth of larger droplets, resulting in a lower number of large cloud droplets (Dp > 18 μm) in the PBL compared to the upper air. $E_d$ in PBL (Fig. 9f, $E_d$ = 12.89 μm) was lower than in the upper air ($E_d$ = 17.94 μm). The inversion layer (Fig.9d, about 750 m in thickness) above the top of PBL enhanced the evaporation activity of cloud droplets, leading to a lower Nc at this height compared to other heights and a higher Na (interstitial aerosol) than that observed at other heights.

Line 383: What is the significance level of the calculated correlation coefficient?

Response: Thank you for your comment. We set the significance level α at 0.05, and a P-value < 0.05 indicates that the result has passed the significance test. We have added this information in lines 409-411 to ensure that our analysis is rigorous.

The equation in the panel represented a fitted curve for the data, indicating the relationship between Na and $E_d$. The relationship between Na and $E_d$ can be expressed

as $E_d = Na^{FIE}$.

Line 388: meteorological parameters instead of 'meteorological elements'

Response: Thank you for your comment. We have changed "meteorological element" to "meteorological parameters."

Line 389: The results in this study are not diurnal since night time observations are missing.

Response: Thank you for your comment. We have revised the content related to "diurnal variabilities" in the manuscript.

Line 394-395: 'Between 1500 m and 3300 m, Na was low, Nc< 200 cm-3, and did not change with the height.' This is not academic writing.

Response: Thank you for your comment. We have changed the statement to "Between 1500 m and 3300 m, the value of Na was low, with Nc remaining below 200 cm⁻³ and not changing with height" in lines 422-424.

Line 401-403: Again, the results are not diurnal.

Response: Thank you for your comment. We have revised the content related to "diurnal variabilities" in the manuscript.

Figures and Tables:

Table 1: What do the values in brackets mean? Are these concentration values corrected to standard conditions?

Response: Thank you for your comment. We have added an explanation for the data in Table 1. The values of Na, Nc, LWC, and $E_d$ in the table are all averages, with their standard deviations.

Figure 1: The results in Figure 1d were not discussed or referred in the text. I am wondering the point of discussing the average values since more specific results as a function of altitude were provided in Figure 1. Given the large spread of the results in Figure 1, an average value of a selected range does not really make sense.

Response: Thank you for your comment. We have re-analyzed the content of Fig.1. Calculating the overall average of the data allows us to identify the trend of each physical quantity with height. This preliminary analysis helps to explore the potential causes and influencing factors affecting the microphysical properties of

clouds in Guangxi.

Figure 2: What is the time zone of the hours? Why the results at 19:00 is missing?

Response: Thank you for your comment. The aircraft did not conduct cloud penetration detection during the 19:00 flight, so there is no data for that time.

Figure 3: The same comments as those on Figure 2. The legend and axis labels can be better organized to improve its readability. Figure 4: The same comments as those on Figure 3. Figure 5: What is the time zone of the hours? What does the x-axis mean? Figure 7: What does the y-axis (Counts) mean? There is no definition in the manuscript. Figure 8: The unit for the colour map scale is missing.

Response: Thank you for your comment. We have added legends and axis labels to the figures to enhance their readability.

[Figure]

**Fig. 2** Vertical profiles of cloud interstitial aerosol concentration, outside aerosol number concentration, and cloud droplet concentration at different times (a is 10:00, b is 11:00, c is 12:00, d is 13:00, e is 14:00, f is 15:00, g is 16:00, h is 17:00, i is 18:00, j is 20:00, the black dashed line represents the height of PBL)

Figure 10: Which kind of correlation coefficient is presented in this Figure? How was it calculated? What is the significance level (p value) of the correlation coefficient? The maximum n value is only 7. Is the equation in each panel representing the

function to calculate the correlation coefficient?

Response: The equation in the panel represents a fitted curve for the data, indicating the relationship between Na and Ed. The relationship between Na and Ed can be expressed as $Ed = Na^{FIE}$. Within different ranges of LWC, FIE is consistently less than 0, indicating that the relationship between aerosol and cloud in the Guangxi region conforms to the Twomey effect. The $R^2$ value reflects the strength of the relationship between the fitted curve and the actual data. We conducted a significance test on the fitting results, setting the significance level $\alpha$ at 0.05 and obtained a P-value < 0.05.

---

## Author Comment (AC3)

Aircraft observation is the most important way to directly obtain the cloud microphysical characteristics. However, due to limited conditions (instruments and funds), there are relatively few aircraft observation data on cloud microphysical characteristics. The aircraft observations of cloud microphysical characteristics in China are mostly concentrated in the Beijing-Tianjin-Hebei region, and related research on southern China is still very rare. Guangxi, as a typical province in southern China, has significant differences in climate characteristics and source emissions from the central and eastern of China. Therefore, the observation of cloud microphysical characteristics in Guangxi is of great significance for a comprehensive understanding of aerosol-cloud interactions in the East Asia. This study gives the data of 9 cloud-crossing aircraft observations, which can provide valuable data support for our in-depth understanding of aerosol-cloud interactions. More importantly, this paper gives the vertical distribution and interaction relationship of aerosols and clouds at different times of the day. Although there is a lack of aircraft observation data at nighttime, the vertical distribution of aerosols and clouds changes little at nighttime due to relatively stable atmospheric conditions, that is, the planetary boundary layer is relatively stable at nighttime and its impact is relatively small. The diurnal variations of the vertical distribution of aerosols and clouds, especially the discussion on the impact of diurnal variations in the planetary boundary layer on their vertical distribution and interaction, are the highlights of this paper. Additionally, this paper discussed the differences in the impact of aerosols from different sources on cloud microphysical properties. They demonstrated that this region's correlation between aerosols and clouds conforms to the Twomey effect. The ultimate goal is to provide observational constraints for the simulation of aerosol radiative forcing in global climate models. Therefore, I recommend that the article should be published after revision.

Thank you for your decision and constructive comments on our manuscript. We have carefully considered the suggestion and tried our best to improve and make changes to the manuscript.

The blue part has been revised according to your comments. The line numbers are in the revised manuscript; the changes are identified in red. Revision notes, point-to-point, are given as follows:

1. There are some formatting errors in the manuscript that need to be carefully revised, such as the lack of space in line 19 and the lack of superscript in line 145. I will not list them one by one, and the author needs to carefully check and revise the entire manuscript.

Response: Thank you for your comment. We have carefully reviewed and corrected the manuscript's formatting errors.

2. abstract, line 37-38, the specific response characteristics of the Twomey effect of aerosol-cloud should be discussed in detail. On the contrary, the observation overview of line 16-21 needs to be simplified and a brief description is sufficient. The direct verification of the Twomey effect through aircraft observation is a very important observational fact and needs to be described in detail.

Response: Thank you for your comment. We have revised the content of the abstract. Based on aircraft observations, we simplified the observation overview and added a description of the specific response characteristics of the Twomey effect of aerosol-cloud. Below is the modified abstract.

Aerosols and clouds play essential roles in the global climate system, and aerosol-cloud interactions significantly impact the earth-atmosphere system's radiation balance, water cycle, and energy cycle. To understand the effect of aerosols on the vertical distribution of stratocumulus microphysical quantities in southwest China, we analyzed data from nine aircraft observations over Guangxi from October 10 to November 3, 2020. This analysis focused on the temporal variation characteristics and formation mechanisms of stratocumulus microphysical profiles, considering the influence of aerosol number concentration on the source of air mass and individual cases. Aerosol number concentration (Na) and cloud droplet concentration (Nc) decreased gradually with the altitude increase below 1500 m and did not change with the height between 1500 m and 3300 m. The temperature inversion layer at the top of the planetary boundary layer (PBL) hindered the increase

in the cloud droplet particle size. The lower layer of the stratocumulus cloud in Guangxi mainly contained small-sized cloud droplets (effective diameter of cloud droplet, $E_d$ < 15 μm), and the middle and upper layer cloud droplet was large particle-size cloud droplet ($E_d$ > 20 μm). The vertical distribution of cloud microphysical quantity had apparent temporal variation. When aerosols in PBL were transported to the upper air (14:00 to 20:00), Nc in the lower layer decreased, and the small particle-size cloud droplets ($E_d$ < 20 μm) in the middle layer and upper layer increased. Aerosols from the free atmosphere were transported into PBL (10:00 to 13:00), providing an abundance of cloud condensation nuclei, which increased the number of small particle-size cloud droplets in the lower layer of the cloud (near the top of PBL). The characteristics of cloud microphysical quantity were also affected by the source of air mass and the height of PBL. Na and Nc were high under the influence of land air mass or aerosols within PBL, and the cloud droplet number concentration spectrum was unimodal. Na and Nc were low under the influence of marine air mass or above the boundary layer, and the cloud droplet number concentration spectrum was bimodal. The relationship between stratocumulus and aerosol in this region is consistent with the Twomey effect. $E_d$ and Na remain negatively correlated in different liquid water content ranges, and FIE (the aerosol first indirect effect) ranged from -0.07 to -0.58.

3. Line 113-116 does not need to be divided into a separate paragraph and can be directly merged with the previous paragraph.

Response: Thank you for your comment. We have merged the content with the previous paragraph to ensure the coherence and flow of the manuscript.

4. 2.1. Aircraft data and reanalyze data. In this section the aircraft data and the reanalysis data are best presented separately.

Response: Thank you for your comment. We have adjusted the order of the introductions for the aircraft observation data and the reanalysis data. Section 2.1 now covers the introduction and processing methods of the aircraft data, while Section 2.2 introduces the reanalysis data. This ensures that readers can more clearly understand the various data sources.

5. line 164. Need to give a reason why Zn is defined?

Response: Thank you for your comment. We have incorporated Zn's necessary information and references in lines 164-167 to address these omissions.

Define the relative heights of the cloud as Zn:

$$Zn = \frac{Z - Z_{base}}{Z_{top} - Z_{base}}$$

In the formula, $Z_{base}$ is the height of the cloud base, and $Z_{top}$ is the height of the cloud top. The cloud heights have been normalized by setting the cloud base as 0 and the cloud top as 1.

6. 3.2 Diurnal variation of the vertical distribution of cloud microphysical quantities. This section is the research highlight of this paper, but the title does not highlight of the research. I recommended to highlight the impact of planetary boundary layer evolution in the title and analysis content. Since this manuscript lacks nighttime observation data, it cannot be strictly called diurnal variation. I recommended that the authors weaken the description of diurnal variation.

Response: Thank you for your comment. We acknowledge that the lack of nighttime observations limits our ability to capture regional diurnal variations in stratocumulus clouds adequately. Our study is based on observations made by nine aircraft during the day, when planetary boundary layer height changes significantly and the vertical transport activity of aerosols is intense, leading to a more pronounced and rapid impact on clouds. We recognize that including nighttime data would provide a more complete picture of these clouds. We have ensured that this limitation is clearly stated in the manuscript and emphasized the impact of planetary boundary layer evolution on cloud microphysical properties. The manuscript's expression "diurnal variation" has been changed to "temporal variation."

7. 3.3 Influence of aerosols on microphysical quantities of stratified clouds. The title of this section directly points to the impact of the Twomey effect.

Response: Thank you for your comment. We have changed the title of Section 3.3 to "Verification of the Twomey Effect" to ensure that it intuitively reflects the content of this section.

---

## Author Response (AR2)

Dear Sergio:

Thank you for your valuable feedback. We have carefully made the following minor and technical corrections. We believe these revisions have enhanced the clarity and quality of our manuscript. Thank you once again for your guidance.

The sections highlighted in blue have been revised according to your comments. The line numbers are indicated in the revised manuscript, and the changes are marked in red. Below are the revision notes, addressed point by point:

point-01

In some figures the color of the points of the legend (indicating Nc and Na, red, black and blue) is too small, is almost not possible to see it unless you zoom it, so please enlarge them. This is the case of almost all figures, but specially in fig. 2, 3, 4, 9

Response: Thank you for your comment. We have enlarged the colors of the points indicating Nc and Na (red, black, and blue) in all figures, particularly in the legends of Fig 1, 2, 3, 4, 6, and 9, to ensure they are more visible and easily distinguishable. The figures in the manuscript have been replaced.

[Figure]

**Fig. 2** Vertical profiles of cloud interstitial aerosol concentration, outside aerosol number concentration, and cloud droplet concentration at different times (a is 10:00, b is 11:00, c is 12:00, d is 13:00, e is 14:00, f is 15:00, g is

16:00, h is 17:00, i is 18:00, j is 20:00, the black dashed line represents the height of PBL)

point-02

Reviewer 1 raised some questions that are properly replied in your report, however such reply did not result in any change in the manuscript, so the future readers of the article may have the same doubt or question than the reviewer. It would be interesting that the information you give in your reply to the reviewer be introduced in the revised version of the article.

Response: Thank you for your comment. We have included relevant clarifications and explanations in the revised version of our manuscript. This addition aims to address potential questions from readers and enhance the overall clarity of our work. In addition, we re-checked our response to reviewer 1's comments and have revised all the contents that "the reply did not result in any change in the manuscript".

Some examples (Ex):

Ex-1.

Reviewer 1 asked (page 4 of the reply report) if the particle diameter is optical or aerodynamic. You replied that is optical, however such information was not introduced in the revised version of the manuscript, and should be included.

Response: Thank you for your comment. We have ensured to explicitly state that the particle diameter mentioned refers to the optical diameter in line 132 and lines 135-136.

A passive cavity aerosol spectrometer probe (PCASP-100X, DMT Inc, USA) was installed to provide aerosol number concentrations in the optical particle size range of 0.11 to 3 μm, with a time resolution of 1s, particle size uncertainty of 20%, and concentration uncertainty of 16%.

Its principle is to detect particles with an optical diameter ranging from 2 μm to 50 μm using forward scattering technology with a time resolution of 1s.

Ex-2.

Reviewer 1 asked (page 5 of the reply report) if the concentrations are in volumetric units or to standards conditions. You replied that in volumetric units, however such information was not been included in the manuscript, and should be included.

Response: Thank you for your comment. We have added the units for Na and Nc in lines 138-139.

Na and Nc are measured in volume units.

Ex-3.

Reviewer 1 comment: Line 28: Does "cloud microphysical quantity" refer to Nc and $E_d$?

Response: Thank you for your comment. In our manuscript, the term "cloud microphysical quantity" indeed refers to both the cloud droplet number concentration (Nc) and the cloud droplet effective diameter ($E_d$).

However, in the revised version of the manuscript (line 28, abstract), the term "cloud microphysical quantity" is still there without clarifying that it is referring to Nc and $E_d$. The reply to the reviewer is clear, but it would require a slight modification in the text to let clear to the reader to what parameters the authors are referring to (i.e. Nc and $E_d$).

Response: Thank you for your comment. We have added an explanation of the cloud microphysical quantity in lines 36-38.

The characteristics of cloud microphysical quantities (Nc and $E_d$) were also affected by the source of air mass and the height of PBL.

Ex-4.

Reviewer 1 asked (page 12 of the reply report) about the use of the term. In the report you replied that the text was modified, and that was replaced by, however, in the revised version of the manuscript with the changes tracked the term has not been replaced (line 94). This should be clarified.

Response: Thank you for your comment. We have updated lines 96-100 in the revised manuscript. Thank you for bringing this to our attention, and we apologize for any confusion caused.

Lu et al. (2007) compared the microphysical quantities of stratocumulus clouds influenced by aircraft flight tracks and those in undisturbed regions and found that the effective radius of cloud droplets in the flight path region was smaller, the number concentration of cloud drops was lower, and the cloud LWC was larger, providing

observational evidence for the first indirect effect of aerosols.

Ex-5.

Reviewer 1 asked (page 13 of the reply report) the correct the grammar of the sentence (We demonstrated that this region's correlation⋯), you replied with a corrected text, however in the version of the manuscript with the changes tracked still includes the old version of the text (lines 116-118) (incorrect grammar). This needs to be corrected or clarified. The reply and the revised version of the manuscript doesn't match.

Response: Thank you for your comment. We have updated lines 121-122 in the revised manuscript. We have checked the manuscript content to ensure that the responses are consistent with the revised version.

Our findings indicate that the interaction between aerosols and clouds in this region aligns with the Twomey effect.

Other comment:

The change of the term "diurnal variations" to "temporal variations" is OK.

You can also use the term "daylight variations", which would be more specific for your study; it would also be OK from my point of view. This is just a suggestion, is optional for you and the other authors.

Response: Thank you for your comment. We have decided to use 'daylight variations' to provide greater specificity in our study.